# Identification of a novel PPARβ/δ/miR-21-3p axis in UV-induced skin inflammation

Gwendoline Degueurce[1], Ilenia D'Errico[1], Christine Pich[1], Mark Ibberson[2], Frédéric Schütz[1,2], Alexandra Montagner[3], Marie Sgandurra[1], Lionel Mury[1], Paris Jafari[4], Akash Boda[1], Julien Meunier[1], Roger Rezzonico[5], Nicolò Costantino Brembilla[6,7], Daniel Hohl[8], Antonios Kolios[9,10], Günther Hofbauer[10], Ioannis Xenarios[2] & Liliane Michalik[1,*]

## Abstract

Although excessive exposure to UV is widely recognized as a major factor leading to skin perturbations and cancer, the complex mechanisms underlying inflammatory skin disorders resulting from UV exposure remain incompletely characterized. The nuclear hormone receptor PPARβ/δ is known to control mouse cutaneous repair and UV-induced skin cancer development. Here, we describe a novel PPARβ/δ-dependent molecular cascade involving TGFβ1 and miR-21-3p, which is activated in the epidermis in response to UV exposure. We establish that the passenger miRNA miR-21-3p, that we identify as a novel UV-induced miRNA in the epidermis, plays a pro-inflammatory function in keratinocytes and that its high level of expression in human skin is associated with psoriasis and squamous cell carcinomas. Finally, we provide evidence that inhibition of miR-21-3p reduces UV-induced cutaneous inflammation in *ex vivo* human skin biopsies, thereby underlining the clinical relevance of miRNA-based topical therapies for cutaneous disorders.

**Keywords** inflammation; miRNA; PPARβ/δ; skin; therapeutics
**Subject Categories** Immunology; Pharmacology & Drug Discovery

## Introduction

Skin disorders accompanied by acute or chronic inflammation are the most common dermatological pathologies. They may be associated with genetic traits (Ellinghaus *et al*, 2013) or environmental factors, such as invading microbial pathogens or solar ultraviolet radiation (UV) (Matsumura & Ananthaswamy, 2004). Although natural sunlight or UV exposure has beneficial aspects—for example, vitamin D production or improved dermatoses (Holick, 2008; Kochevar *et al*, 2008)—excessive exposure to UV is widely recognized as a major factor leading to skin perturbations (Kochevar *et al*, 2008). The skin's response to UV exposure includes inflammation, disruption of the epidermal barrier function, premature aging, and ultimately, UV-induced carcinogenesis (Holleran *et al*, 1997; Baumann, 2007; Narayanan *et al*, 2010; Biniek *et al*, 2012). Therefore, it is important to further understand the complex and incompletely characterized mechanisms underlying inflammatory skin disorders resulting from UV exposure.

The nuclear hormone receptor peroxisome proliferator-activated receptor β/δ (PPARβ/δ)—the prevalent PPAR subtype in human and murine epidermis—is an important player in the maintenance of skin homeostasis. It regulates keratinocyte differentiation and lipid synthesis, restores epidermal barrier function following a mechanical disruption, and attenuates UVB-induced senescence in keratinocytes (Michalik & Wahli, 2007; Pal *et al*, 2011; Sertznig & Reichrath, 2011; Ham *et al*, 2012). Upon skin injury, PPARβ/δ activation promotes skin healing through activation of keratinocyte proliferation, survival, and migration (Michalik *et al*, 2001; Tan *et al*, 2001, 2005). In contrast to these beneficial functions, we recently showed that epidermal activation of PPARβ/δ also favored the progression of UV-induced skin squamous cell carcinoma (Montagner *et al*, 2013). The outcome of PPARβ/δ activity essentially relies on transcriptional activation of target genes, but also on less frequent indirect repressive effects (Feige *et al*, 2006). Although PPARs were recently reported to regulate the expression of a few miRNAs (Shah *et al*, 2007; Yin *et al*, 2010; Gan *et al*, 2013; Song

1  Center for Integrative Genomics, Faculty of Biology and Medicine, University of Lausanne, Lausanne, Switzerland
2  SIB Swiss Institute of Bioinformatics, University of Lausanne, Lausanne, Switzerland
3  INRA ToxAlim, Integrative Toxicology and Metabolism, UMR1331, Toulouse, France
4  Department of Musculoskeletal Medicine, Service of Plastic and Reconstructive Surgery, CHUV, Epalinges, Switzerland
5  Institut de Pharmacologie Moléculaire et Cellulaire, CNRS, UMR 7275, Valbonne, France
6  Dermatology, University Hospital and School of Medicine, Geneva, Switzerland
7  Immunology and Allergy, University Hospital and School of Medicine, Geneva, Switzerland
8  Service de dermatologie et venereology, Hôpital de Beaumont, CHUV, Lausanne, Switzerland
9  Department of Immunology, University Hospital, University of Zürich, Zürich, Switzerland
10 Department of Dermatology, University Hospital, University of Zürich, Zürich, Switzerland
   *Corresponding author. Tel: +41 21 692 41 10; Fax: +41 21 692 41 15; E-mail: liliane.michalik@unil.ch

*et al*, 2013; Panza *et al*, 2014; Yu *et al*, 2014; Dharap *et al*, 2015), the underlying mechanisms and outcomes are poorly understood.

The role of miRNAs in the skin was first demonstrated by the conditional inactivation of the small RNA-processing pathway in keratinocytes, which resulted in striking defects in skin morphogenesis (Andl *et al*, 2006; Yi *et al*, 2006, 2009). Since then, miRNAs—among them the oncomiR miR-21-5p, one of the most studied miRNAs—have been shown to be implicated in the regulation of skin homeostasis (Yi *et al*, 2006, 2008), skin repair (Banerjee *et al*, 2011; Yang *et al*, 2011; Pastar *et al*, 2012; Li *et al*, 2015), and skin disorders such as psoriasis (Joyce *et al*, 2011; Xia & Zhang, 2014; Hawkes *et al*, 2016) and squamous cell carcinomas (Medina *et al*, 2010; Darido *et al*, 2011; Lefort *et al*, 2013; Gastaldi *et al*, 2014). Recent studies indicate that miRNAs are affected by UV exposure in various isolated cell types (Guo *et al*, 2009; Pothof *et al*, 2009; Dziunycz *et al*, 2010; Hou *et al*, 2013), but their functions in the skin response to UV still remain to be characterized.

Here, we identify miR-21-3p as a UV- and PPARβ/δ-activated pro-inflammatory miRNA in keratinocytes in culture and *in vivo*, and we propose that topical inhibition of miR-21-3p is of therapeutic interest in human inflammatory skin disorders.

## Results

### miR-21-3p is an epidermal, UV-induced, PPARβ/δ-activated miRNA

We recently demonstrated that Ppard$^{+/+}$ mice chronically exposed to ultraviolet radiation (UV) displayed earlier skin lesions and faster progression of UV-induced skin carcinogenesis compared to Ppard$^{-/-}$ animals (Montagner *et al*, 2013). In order to identify PPARβ/δ-regulated miRNA involved in the skin response to UV, we compared miRNA expression in skin samples harvested from Ppard$^{+/+}$ and Ppard$^{-/-}$ mice non-irradiated (control), or following acute (24 h after a single dose of UV) or chronic UV exposure (12 weeks of repeated UV exposure; non-lesional skin). This comparative study highlighted 12 major miRNAs, whose expression was affected ($\geq$ 1.5-fold) in a PPARβ/δ-dependent manner, among which nine

were overexpressed, while the remaining three showed lower levels in Ppard$^{+/+}$ compared to Ppard$^{-/-}$ skin (Appendix Table S1). Among the miRNAs whose expression was upregulated by UV exposure in Ppard$^{+/+}$ skin, miR-21-3p particularly attracted our attention as it is the passenger miRNA of the guide miR-21-5p (commonly named miR-21). MiR-21-5p is a well-characterized "oncomiR" induced by UV irradiation (Guo *et al*, 2009; Hou *et al*, 2013) and known for its oncogenic role in skin squamous cell carcinomas (Darido *et al*, 2011; Xu *et al*, 2012; Bruegger *et al*, 2013). Although passenger miRNAs are commonly thought to be degraded upon miRNA processing, here we confirmed miR-21-3p expression by quantifying RNA sequencing counts in various murine organs, including the skin (Meunier *et al*, 2013). The increased miR-21-3p/miR-21-5p ratio in the skin compared to other organs (Appendix Fig S1A) was not due to a lower expression in miR-21-5p, but to enrichment in miR-21-3p expression in that organ (Fig 1A). *In situ* hybridization performed in Ppard$^{+/+}$ skin revealed that miR-21-3p was expressed in the epidermis and hair follicles, with little or no expression in the dermis (Fig 1B, top left panel). Following acute UV exposure, miR-21-3p level was strongly increased in Ppard$^{+/+}$ epidermis, while remaining below detection levels in the dermis (Fig 1B, bottom left panel). We confirmed and quantified the epidermal increase of miR-21-3p expression following UV exposure using RT–qPCR (Fig 1C) and RNA sequencing (Appendix Fig S1B) of isolated epidermis and dermis samples, whose successful separation was confirmed using specific markers (Appendix Fig S1C). Notably, *in vivo* miR-21-3p localization and expression compare with those of PPARβ/δ mRNA, also upregulated in the epidermis upon UV exposure (Appendix Fig S1D).

PPARβ/δ-dependent upregulation of miR-21-3p was then demonstrated in models of genetic and pharmacological modulation of PPARβ/δ function. *In situ* hybridization and RT–PCR quantification revealed that while miR-21-3p level was upregulated in Ppard$^{+/+}$ skin samples in response to acute and chronic UV exposure, it remained expressed at its basal level in the skin of Ppard$^{-/-}$ animals (Fig 1B–D). Moreover, *in vivo* topical inhibition of PPARβ/δ with an antagonist significantly reduced the magnitude of miR-21-3p UV-dependent increase in the epidermis of Ppard$^{+/+}$ mice, but did not

---

**Figure 1. PPARβ/δ activates the expression of UV-induced epidermal miR-21-3p.**

A   RT–qPCR quantification of relative miR-21-3p and miR-21-5p levels in mouse brain, heart, kidney, and epidermis. *N* = 4 animals per group, one representative experiment is shown out of three independent replicates.

B   Fluorescent miR-21-3p *in situ* hybridization (pink) in dorsal skin of acutely irradiated (Ac-UV) and non-irradiated (no UV) Ppard$^{+/+}$ and Ppard$^{-/-}$ mice. E: epidermis; D: dermis; HF: hair follicle. Scale bar: 100 μm.

C   RT–qPCR quantification of relative pri-miR-21 and miR-21-3p levels in the epidermis (left and middle) and of relative miR-21-3p level in the dermis (right) of acutely irradiated (Ac-UV; +) and non-irradiated (−) Ppard$^{+/+}$ and Ppard$^{-/-}$ mice. Pri-miR-21: Ppard$^{+/+}$ Ac-UV vs. Ppard$^{-/-}$ Ac-UV *P* = 0.022; miR-21-3p: Ppard$^{+/+}$ no UV vs. Ppard$^{+/+}$ Ac-UV *P* = 0.017, Ppard$^{+/+}$ Ac-UV vs. Ppard$^{-/-}$ Ac-UV *P* = 0.008. *N* = 3–4 animals per group, one representative experiment is shown out of three independent replicates.

D   RT–qPCR quantification of relative miR-21-3p levels in total skin of chronically irradiated (Chr-UV; +) and non-irradiated (−) Ppard$^{+/+}$ and Ppard$^{-/-}$ mice. miR-21-3p: Ppard$^{+/+}$ no UV vs. Ppard$^{+/+}$ Chr-UV *P* = 0.008, Ppard$^{+/+}$ Chr-UV vs. Ppard$^{-/-}$ Chr-UV *P* = 0.002, ns: non-significant. *N* = 4 animals per groups, one representative experiment is shown out of two independent replicates.

E   RT–qPCR quantification of relative miR-21-3p level in the epidermis of Ppard$^{+/+}$ and Ppard$^{-/-}$ mice, acutely irradiated (Ac-UV; +) or non-irradiated (−), treated with the PPARβ/δ antagonist GSK0660 (+) or vehicle (−), as indicated. miR-21-3p: Ppard$^{+/+}$ no UV vs. Ppard$^{+/+}$ Ac-UV *P* = 0.005, Ppard$^{+/+}$ Ac-UV vs. Ppard$^{+/+}$ Ac-UV/GSK0660 *P* = 0.04, *N* = 5 (Ppard$^{+/+}$) to 3 (Ppard$^{-/-}$) animals per group. One representative experiment is shown out of two independent replicates

F   RT–qPCR quantification of relative miR-21-3p levels in HaCat cells treated with the PPARβ/δ agonists GW501516, GW0742 (+), or vehicle (−) as indicated. miR-21-3p: Veh. vs. GW501516 *P* = 4E-04, Veh. vs. GW0742 *P* = 2E-04. *N* = 2–3 biological replicates, one representative experiment is shown out of two independent replicates.

Data information: Results are presented as mean values ± SEM. The statistical comparison between groups was performed by using *t*-test. \**P*-value < 0.05; \*\**P*-value < 0.01.

affect miR-21-3p expression in the epidermis of Ppard$^{-/-}$ mice (Fig 1E). Finally, the upregulation of the human miR-21-3p by PPARβ/δ was also confirmed in the human keratinocytes HaCaT following activation of PPARβ/δ with its two agonists GW501516 and GW0742 (Fig 1F), like the two well-characterized PPARβ/δ target genes Angptl4 and Tgfb1 (Appendix Fig S1E).

Collectively, these findings establish that the passenger miRNA miR-21-3p is selectively expressed in the epidermis where it is strongly upregulated in response to UV exposure and that PPARβ/δ is an activator of both murine and human miR-21-3p.

## PPARβ/δ activates miR-21-3p expression indirectly via TGFβ1

The gene encoding miR-21-3p and miR-21-5p (MIR21) is transcribed into the primary transcript pri-miR-21, which is further processed into pre-miR-21. Pre-miRNA is in turn processed into a duplex consisting of the passenger miR-21-3p and the guide miR-21-5p by the Dicer complex (Mah *et al*, 2010; Kumarswamy *et al*, 2011). UV-induced expression of pri-miR-21 was partially but significantly reduced in Ppard$^{-/-}$ compared to Ppard$^{+/+}$ epidermis (Fig 1C, left panel), and activation of PPARβ/δ with its agonist GW0742 resulted

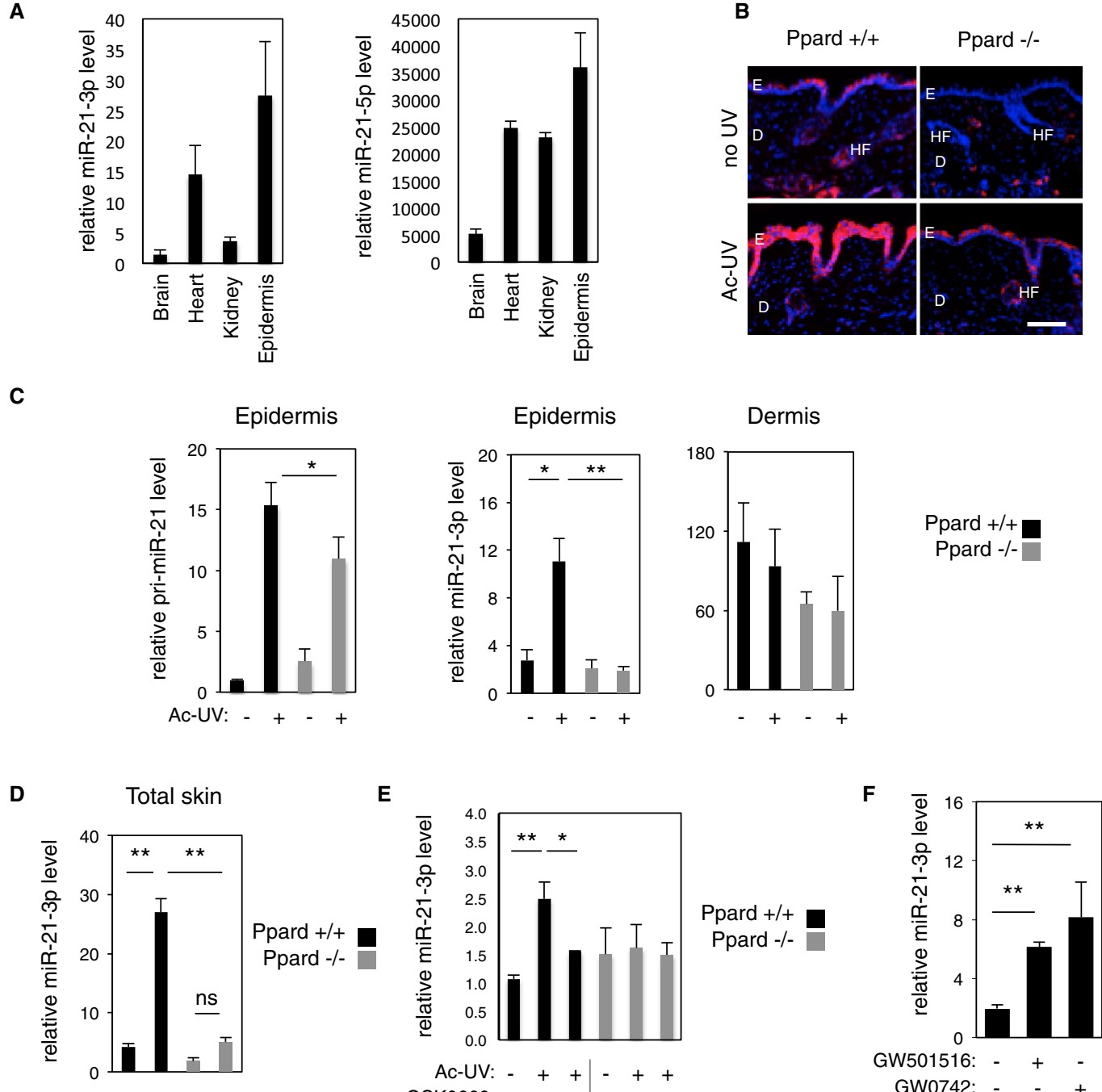

Figure 1.

in an increase of pri-miR-21 expression in human keratinocytes (Fig 2A), suggesting a transcriptional regulation of the miR-21-5p/miR-21-3p-encoding gene by PPARβ/δ. However, *in silico* analyses did not reveal any PPAR binding site (PPAR response elements, direct repeats of DR1 type) in the promoter of MIR21 (Ribas *et al*, 2012). Furthermore, the inhibition of protein synthesis with cycloheximide completely prevented PPARβ/δ-dependent upregulation of pri-miR-21 in human keratinocytes (Fig 2A). These data suggest that PPARβ/δ activates the transcription of the miR-21-5p/miR-21-3p encoding gene in an indirect fashion.

Tgfb1 is a well-characterized PPARβ/δ direct target gene (Kim *et al*, 2008; Montagner *et al*, 2013), and TGFβ1 signaling was shown to activate the transcription of miR-21-5p in kidney cells (Godwin *et al*, 2010; Zhong *et al*, 2011). Thus, we addressed the hypothesis that PPARβ/δ indirectly upregulated the expression of the miR-21-3p/miR-21-5p encoding gene through activation of its direct target gene Tgfb1. We first showed that pri-miR-21, pre-miR-21, miR-21-3p, and miR-21-5p were upregulated by TGFβ1 in human keratinocytes (Fig 2B), using the TGFβ1 target gene SERPINE1 as a positive control (Appendix Fig S1F). To test whether the transcriptional regulation of the miR-21-3p/miR-21-5p encoding gene by PPARβ/δ was TGFβ1 dependent, we combined PPARβ/δ activation (GW0742) with TGFβ receptor inhibition (SB431542) in human keratinocytes and monitored the expression of ANGPTL4 and SERPINE1—PPARβ/δ and TGFβ1 respective target genes—as controls for treatment efficiency (Appendix Fig S1G). PPARβ/δ activation with its agonist resulted in upregulation of pri-miR-21, pre-miR-21, and miR-21-3p (Fig 2C), which was prevented by the inhibition of the TGFβ receptor. While miR-21-5p level was downregulated by TGFβ receptor inhibition, it was not significantly affected by PPARβ/δ activation (Fig 2C).

We next addressed whether the PPARβ/δ-dependent, UV-induced upregulation of miR-21-3p observed in murine skin *in vivo* also required TGFβ receptor activity. Mice were exposed to a single dose of UV, with or without cutaneous topical application of the TGFβ receptor inhibitor. As expected for a direct PPARβ/δ target gene, we confirmed that Tgfb1 expression was increased by acute UV exposure in Ppard$^{+/+}$ but not in Ppard$^{-/-}$ epidermis (Fig 2D; Montagner *et al*, 2013). The expression levels of miR-21-3p and miR-21-5p were upregulated by UV in Ppard$^{+/+}$ epidermis, an activation that was abolished by topical inhibition of the TGFβ receptor (Fig 2E), as also

observed for the TGFβ1 target gene SERPINE1 used as a positive control (Appendix Fig S1H).

Interestingly, *in silico* analyses to generate a list of predicted miR-21-3p target mRNA using Diana-MicroT-CDS miRNA database (Reczko *et al*, 2012; Paraskevopoulou *et al*, 2013) revealed SMAD7 as a putative direct target of miR-21-3p (by Hits-Clips, according to Tarbase v7.0 (Vergoulis *et al*, 2012)). Among the predicted miR-21-3p targets (Appendix Table S2), SMAD7 was of particular interest as it acts as an antagonist of TGFβ1 signaling (Nakao *et al*, 1997; Yan *et al*, 2015) and its expression is activated by UV in both murine and human skin (Fig 3A; Quan *et al*, 2001). *In silico* sequence analysis using miRmap interface (Vejnar *et al*, 2013) predicted two miR-21-3p binding sites in the human SMAD7 3′UTR, consisting, respectively, of seven and six perfect nucleotide matches (Fig 3B, left). The sequences of both binding sites are 100% conserved between human and mouse. Luciferase reporter assays using the 3′UTR of SMAD7 demonstrated that miR-21-3p mimic delivery significantly reduced the activity of the wild-type SMAD7 3′UTR reporter (44%), but not that of the miR-21-3p binding site mutant reporter (Fig 3B). miR-21-3p mimic delivery to human keratinocytes did not affect endogenous SMAD7 mRNA expression (Fig 3C, left panel), but decreased SMAD7 protein level by 50% (Fig 3C, middle and right panels). Together, these data indicate that SMAD7 is a direct target of miR-21-3p regulated at the translational level. However, expression of miR-21-3p (Fig 1C) and Smad7 (Fig 3D) was not anti-correlated in the epidermis of Pparβ$^{+/+}$ and Pparβ$^{-/-}$ mice exposed to UV, indicating that although miR-21-3p likely contributes to its regulation, mouse Smad7 level is under unsurprising more complex regulation *in vivo*. Of note, miR-21-3p may enhance TGFβ1 signaling via downregulation of SMAD7. Consistent with this hypothesis, the TGFβ1 targets SERPINE, p21 and RUNX were expressed at a higher level, while the expression levels of TGFB1 and its target SNAI2 were not significantly affected, in miR-21-3p mimic-overexpressing HaCat cells following TGFβ1 treatment (Fig 3E).

Taken together, these results suggest that PPARβ/δ indirectly activates the transcription of the gene encoding miR-21-3p likely through increased expression of TGFβ1 and activation of TGFβ receptor. By preventing excessive SMAD7 protein upregulation, miR-21-3p may maintain its own expression by activating TGFβ signaling.

---

**Figure 2.   Activation of miR-21-3p by PPARβ/δ requires activation of the TGFβ receptor.**

A   RT–qPCR quantification of relative pri-miR-21 level in HaCaT human keratinocytes treated with the PPARβ/δ agonist GW0742 (+) or vehicle (−), with (+) or without (−) cycloheximide (Cyclo) as indicated. Pri-miR-21: Veh vs. GW0742 *P* = 0.009. *N* = 3 biological replicates, one representative experiment is shown out of two independent replicates.

B   RT–qPCR quantification of relative pri-miR-21, pre-miR-21, miR-21-5p, and miR-21-3p levels in HaCaT cells treated for 24 h with 2 or 5 ng/ml of recombinant human TGFβ1 (+) or vehicle (−) as indicated. Pri-miR-21: Veh vs. TGFβ1 5 ng/ml *P* = 0.029; Pre-miR-21: Veh vs. TGFβ1 5 ng/ml *P* = 0.001; miR-21-5p: Veh vs. TGFβ1 5 ng/ml *P* = 0.002; miR-21-3p: Veh vs. TGFβ1 5 ng/ml *P* = 1.7E-05. *N* = 3 biological replicates, one representative experiment is shown out of two independent replicates.

C   RT–qPCR quantification of pri-miR-21, pre-miR-21, miR-21-5p, and miR-21-3p levels in HaCat cells treated for 24 h with the PPARβ/δ agonist GW0742 (+), TGFβ receptor inhibitor SB431542 (+), or vehicle (−) as indicated. Pri-miR-21: GW0742 vs. SB431542 *P* = 0.004, GW0742 vs. GW0742/SB431542 *P* = 0.015; Pre-miR-21: Veh vs. GW0742 *P* = 0.022, GW0742 vs. SB431542 *P* = 0.036, GW0742 vs. GW0742/SB431542 *P* = 0.024; miR-21-5p: GW0742 vs. SB431542 *P* = 0.034, GW0742 vs. GW0742/SB431542 *P* = 0.024; miR-21-3p: Veh vs. GW0742 *P* = 0.036, GW0742 vs. SB431542 *P* = 0.011, GW0742 vs. GW0742/SB431542 *P* = 0.008. *N* = 3 biological replicates, one representative experiment is shown out of two independent replicates.

D   RT–qPCR quantification of relative Tgfb1 level in the epidermis of acutely irradiated (Ac-UV; +) and non-irradiated (−) Ppard$^{+/+}$ and Ppard$^{-/-}$ mice. Tfgb1: Ppard$^{+/+}$ no UV vs. Ac-UV *P* = 0.034, Ppard$^{+/+}$ Ac-UV vs. Ppard$^{-/-}$ Ac-UV *P* = 0.030. *N* = 2–3 animals per group, one representative experiment is shown out of three independent replicates.

E   RT–qPCR quantification of relative miR-21-5p and miR-21-3p levels in the skin of Ppard$^{+/+}$ mice treated with the TGFβ receptor inhibitor SB431542 (+) or vehicle (−), with (+) or without (−) acute UV exposure (Ac-UV). miR-21-5p: no UV vs. Ac-UV *P* = 0.038, Ac-UV vs. Ac-UV/SB431542 *P* = 0.028; miR-21-3p: no UV vs. Ac-UV *P* = 0.017; Ac-UV vs. Ac-UV/SB431542 *P* = 0.035. *N* = 3 animals per group, one representative experiment is shown out of three independent replicates.

Data information: Results are presented as mean values ± SEM. The statistical comparison between groups was performed by using *t*-test. *\*P*-value < 0.05; *\*\*P*-value < 0.01.

    

## MiR-21-3p is a pro-inflammatory miRNA in keratinocytes

To investigate the role of miR-21-3p, we mimicked UV-induced miR-21-3p upregulation by transfecting human keratinocytes with a miR-21-3p mimic oligonucleotide sequence, using a scrambled sequence as a control (Appendix Fig S2A). The global impact of miR-21-3p gain of function on keratinocyte mRNA expression was investigated by microarray profiling. This analysis revealed that of all the mRNA which expression was affected by miR-21-3p mimic transfection, 33% were associated with inflammatory and immune processes (Fig 4A); the four most affected biological processes were immune response, defense response, response to wounding, and

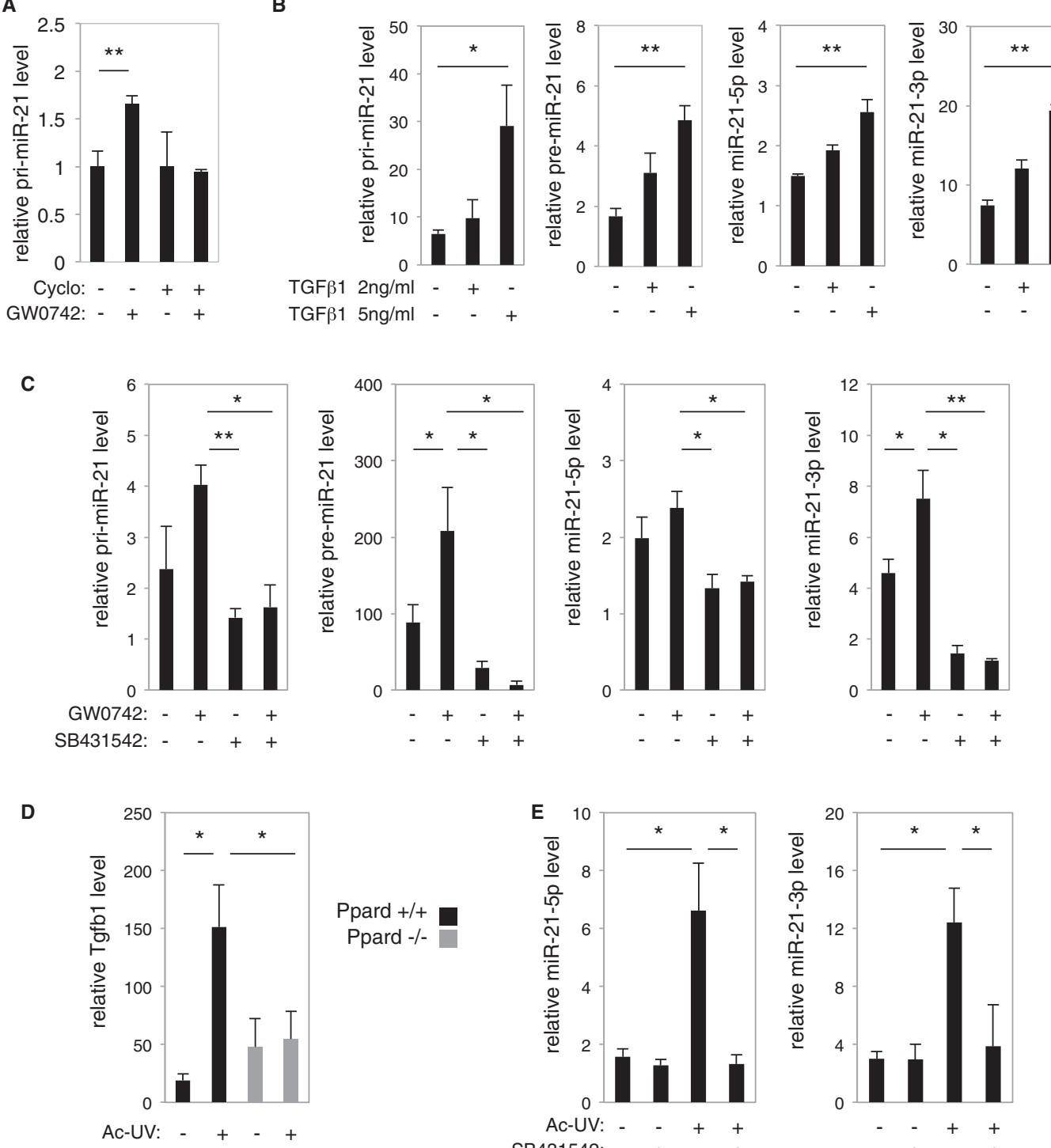

Figure 2.

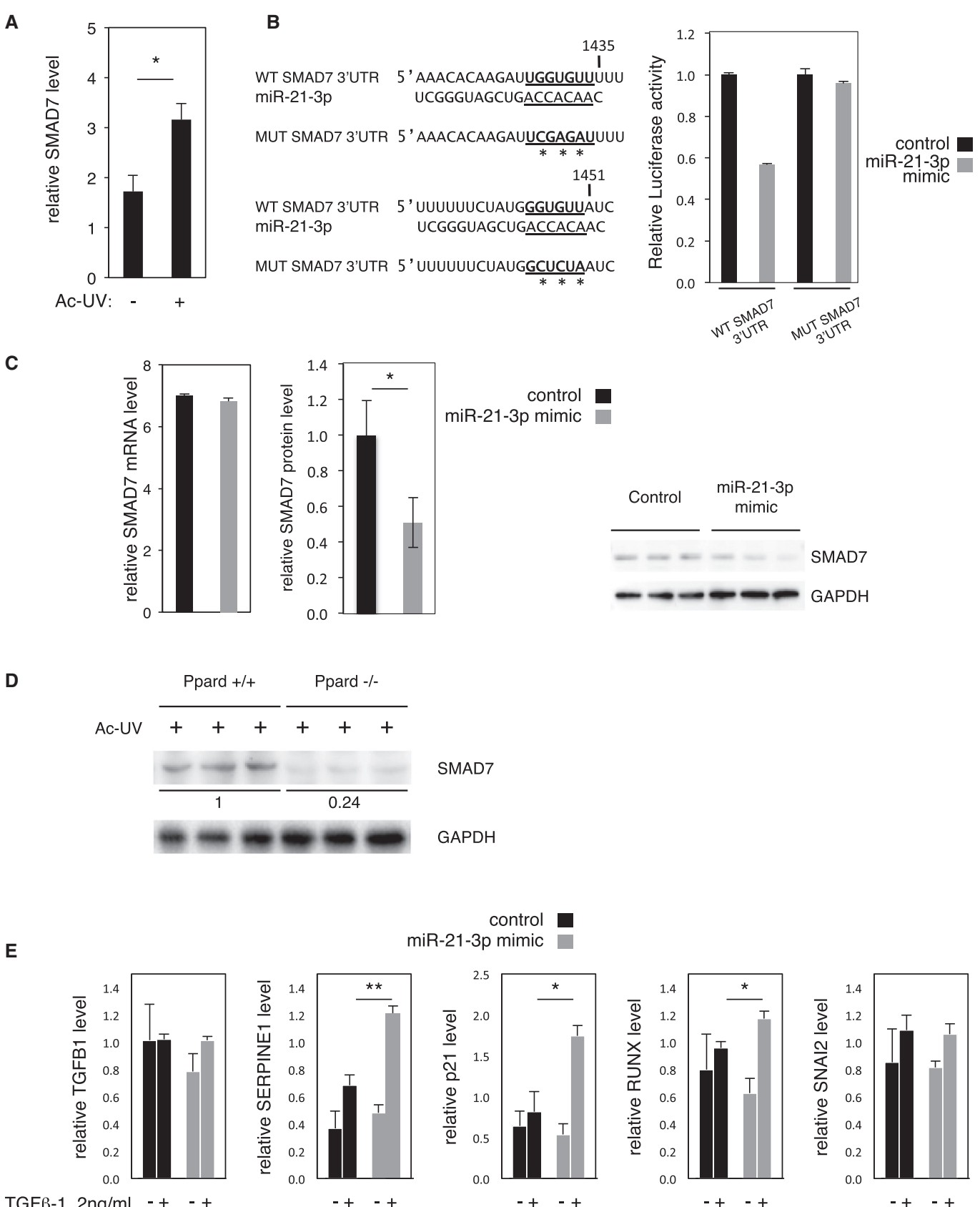

Figure 3.

◄

**Figure 3. SMAD7 is a target for miR-21-3p.**

A   RT–qPCR quantification of relative SMAD7 expression level in the epidermis of acutely irradiated (Ac-UV; +) *ex vivo* biopsies of human normal skin. SMAD7: no UV vs. Ac-UV *P* = 0.032. *N* = 4 biological replicates, one representative experiment is shown out of three independent experiments performed with the skin of three different donors.

B   Left panel: wild-type (WT SMAD7 3′UTR) and mutated (MUT SMAD7 3′UTR; *: mutated nucleotides) miR-21-3p binding sequences in the human SMAD7 3′UTR. Right panel: Luciferase reporter assay with wild-type (WT SMAD7 3′UTR) or mutated (MUT SMAD7 3′UTR) SMAD7 3′UTR in HEK 293 cells overexpressing miR-21-3p (miR-21-3p mimic) or a scrambled sequence (Control).

C   Left panel: normalized expression data of SMAD7 mRNA obtained from genomic microarray analysis of human HaCaT cells treated with a miR-21-3p mimic (miR-21-3p mimic) or a scrambled sequence (control). *N* = 3 biological replicates. Middle panel: Western blot quantification of SMAD7 protein level (normalized to GAPDH protein level) in human HaCaT cells treated with a miR-21-3p mimic (miR-21-3p mimic) or a scrambled sequence (control). SMAD7 protein: control vs. miR-21-3p mimic *P* = 0.031. *N* = 3 biological replicates. Right panel: Western blot of SMAD7 and GAPDH proteins in human HaCaT cells treated with a miR-21-3p mimic (miR-21-3p mimic) or a scrambled sequence (control), *N* = 3 biological replicates; one representative experiment is shown out of two independent replicates.

D   Western blot of Smad7 from epidermis of acutely irradiated (Ac-UV; +) Ppard$^{+/+}$ and Ppard$^{−/−}$ mice. Loading control: GAPDH.

E   RT–qPCR quantification of relative TGFB1, SERPINE1, p21, RUNX, and SNAI2 levels in human HaCaT cells treated with a miR-21-3p mimic (miR-21-3p mimic) or a scrambled sequence (control) with (+) or without (−) treatment with 2 ng/ml recombinant TGFβ1. SERPINE1: TGFβ1 2 ng/ml control vs. miR-21-3p mimic *P* = 0.004; p21: TGFβ1 2 ng/ml control vs. miR-21-3p mimic *P* = 0.03; RUNX: TGFβ1 2 ng/ml control vs. miR-21-3p mimic *P* = 0.04. *N* = 3 biological replicates.

Data information: Results are presented as mean values ± SEM. The statistical comparison between groups was performed by using *t*-test. *$P$-value < 0.05; **$P$-value < 0.01.

inflammatory response (Fig 4B). Among the pro-inflammatory mediators whose level was affected by miR-21-3p in human keratinocytes, we confirmed a strong RNA upregulation of the pro-inflammatory cytokines IL-6 and IL-1B, of the IL-1 receptor co-factor IL-1RAP, of the prostanoid-producing enzyme cyclooxygenase-2 (PTGS2), and of the chemokines CCL5 and CXCL10 (Fig 4C). Interestingly, the expression of caspase-14 (CASP14), whose down-regulation was shown following UV-mediated epidermal barrier disruption, was significantly reduced upon miR-21-3p gain of function (Fig 4C).

Given this pro-inflammatory role of miR-21-3p in keratinocytes, we next analyzed whether the higher levels of miR-21-3p observed in Ppard$^{+/+}$ compared to Ppard$^{−/−}$ epidermis exposed to UV were associated with exacerbated inflammation *in vivo*. Like in human keratinocytes, a higher expression of miR-21-3p was indeed associated with at least double Il6, Il1b, Ptgs2, and Ccl5 mRNA expressions in Ppard$^{+/+}$ compared to Ppard$^{−/−}$ epidermis after acute UV exposure (Fig 5A), and UV-induced recruitment of macrophages was exacerbated in Ppard$^{+/+}$ compared to Ppard$^{−/−}$ dermis (Fig 5B). Moreover, miR-21-3p expression was also increased in other murine skin lesions associated with inflammation, such as chemically induced cutaneous murine papilloma and squamous cell carcinomas (Appendix Fig S2B), while moderate epidermal barrier disruption (obtained by tape stripping and monitored using involucrin as a control) did not cause an increase in miR-21-3p expression (Appendix Fig S2C). Interestingly, subcutaneous delivery of a miR-21-3p inhibitor in mice tended to reduce acute UV-induced inflammation as indicated by a reduction in UV-induced Il6, Ptgs2, and TNF-α levels in the epidermis, although comparison with the control-treated epidermis did not reach statistical significance (Fig 5C).

Collectively, these observations demonstrate that miR-21-3p promotes inflammation in human keratinocytes in culture. Consistently, high expression levels of miR-21-3p are associated with exacerbated inflammation in murine skin exposed to UV or to chemical carcinogens, while miR-21-3p inhibition in mouse skin *in vivo* tends to reduce acute UV-induced inflammation.

### Inhibition of miR-21-3p is of clinical relevance in human skin

We next investigated the relevance of miR-21-3p pro-inflammatory function in inflammatory human skin disorders. We found that, like

in the murine skin, human cutaneous miR-21-3p expression was localized in the epidermis (Appendix Fig S2D). Importantly, miR-21-3p levels were higher in human squamous cell carcinoma (SCC; Fig 6A, right panel) and in human psoriasis lesions (Fig 6B, right panel) compared with healthy human skin. In line with a transcriptional activation of miR-21-3p expression, high level of miR-21-3p in these samples correlated with high levels of pri-miR-21, pre-miR-21, and guide strand miR-21-5p (Fig 6A and B), as well as with high levels of PPARD and TGFB1 mRNAs (Appendix Fig S2E and F). The observation that elevated miR-21-3p was associated with inflammation in murine skin exposed to UV, in human kerati-nocytes, and in human skin with inflammatory disorders raises the exciting possibility that miR-21-3p inhibitors may be used as thera-peutic anti-inflammatory agents. We thus tested the hypothesis that inhibition of miR-21-3p would mitigate the inflammatory response in human skin subjected to acute UV exposure. A miR-21-3p LNA inhibitor or its scrambled control was topically applied on human abdominal skin *ex vivo* explants before and following acute UV exposure. Successful formation of the stable duplex between miR-21-3p and its LNA inhibitor in the epidermis was controlled by the lack of miR-21-3p PCR amplification in the LNA inhibitor-treated samples, while increase in miR-21-3p expression was detected as expected in scrambled-treated samples (Appendix Fig S2G). Treat-ment with the miR-21-3p inhibitor significantly reduced the peaks of IL6, IL1B, and PTGS2 in the human skin explants following UV exposure compared to control-treated biopsies (Fig 6C), while the expression of TGFB1 was unaffected (Appendix Fig S2H).

These findings highlight the pathophysiological relevance of miR-21-3p upregulation in human inflammatory skin disorders and suggest that suppression of miR-21-3p activity may provide a thera-peutic benefit, at least for the prevention of UV-induced skin inflammation.

## Discussion

In this study, we describe a novel molecular cascade, involving a PPARβ/δ- and TGFβ-dependent activation of miR-21-3p, which we identify as a UV-induced, pro-inflammatory miRNA in keratinocytes. This cascade is of pathophysiological importance since miR-21-3p levels are increased in murine and human skin exposed to UV, in human psoriatic skin, and in human squamous cell carcinomas

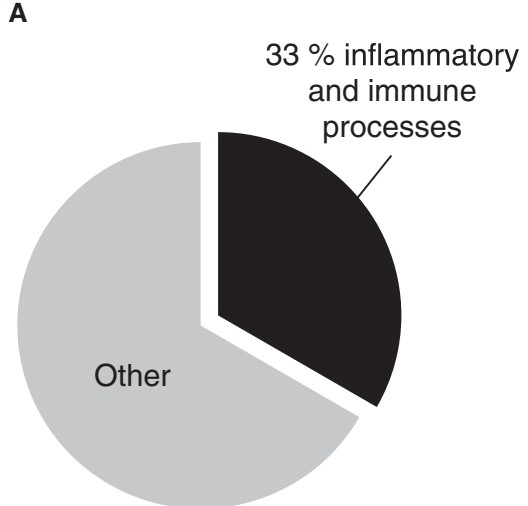

**B**

| Biological Process (GO term) | P-Value |
|---|---|
| Immune response | 2.10E-19 |
| Defense response | 3.10E-08 |
| Response to wounding | 2.70E-05 |
| Inflammatory response | 4.70E-05 |
| Regulation of protein kinase cascade | 5.80E-05 |
| Positive regulation of cell communication | 7.50E-04 |
| Positive regulation of signal transduction | 1.00E-03 |

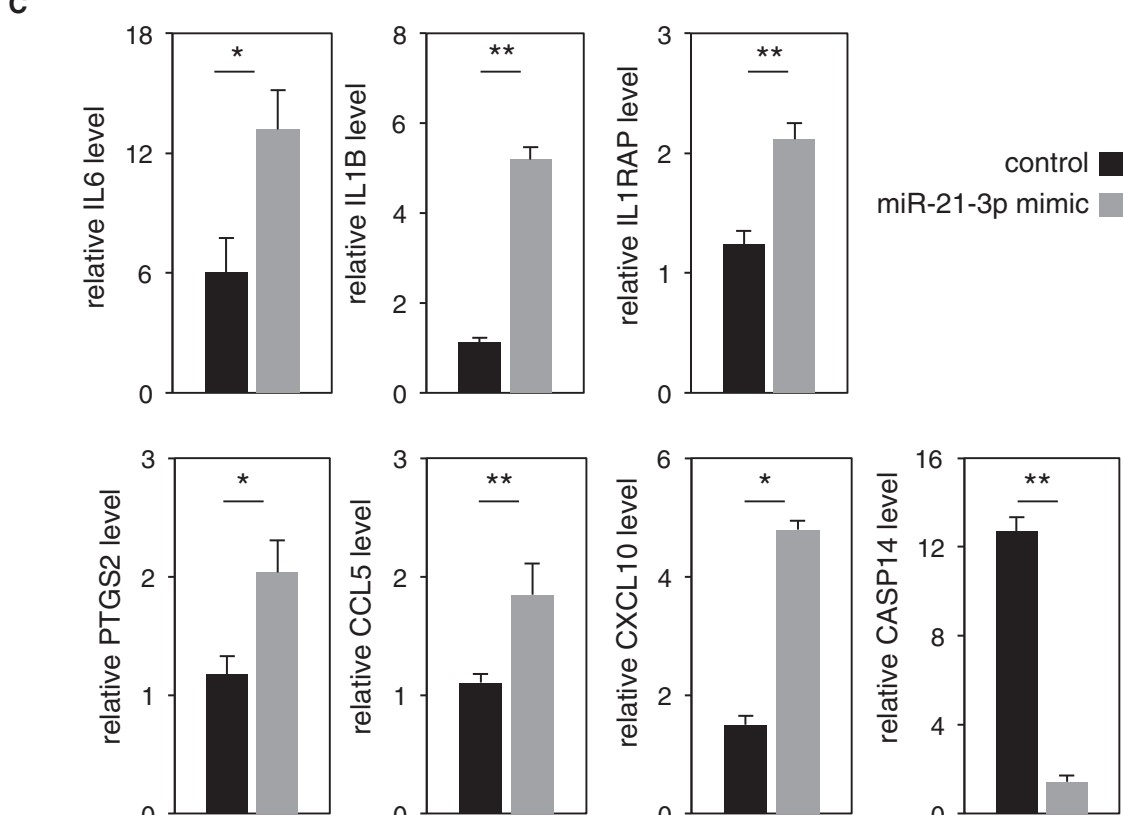

**Figure 4. Gain of miR-21-3p function activates inflammation in HaCaT human keratinocytes.**

A Proportion of inflammatory- and immune processes-associated mRNA in regard to the total number of significantly deregulated mRNA in miR-21-3p mimic versus scrambled control-treated HaCaT cells.

B GO term enrichment analysis based on the significantly upregulated and downregulated mRNA in miR-21-3p mimic-treated compared to scramble control-treated HaCaT cells.

C RT–qPCR analysis of IL6, IL1B, IL1RAP, PTGS2, CCL5, CXCL10, and CASP14 levels in HaCat cells transfected with a miR-21-3p mimic or a scrambled sequence (control). IL6: control vs. miR-21-3p mimic $P = 0.05$; IL1B: control vs. miR-21-3p mimic $P = 0.005$; IL1RAP: control vs. miR-21-3p mimic $P = 0.008$; PTGS2: control vs. miR-21-3p mimic $P = 0.050$; CCL5: control vs. miR-21-3p mimic $P = 0.001$; CXCL10: control vs. miR-21-3p mimic $P = 0.012$; CASP: control vs. miR-21-3p mimic $P = 9E-05$. $N = 3$ biological replicates, one representative experiment is shown out of three independent replicates.

Data information: Results are presented as mean values $\pm$ SEM. The statistical comparison between groups was performed by using $t$-test. *$P$-value $< 0.05$; **$P$-value $< 0.01$.

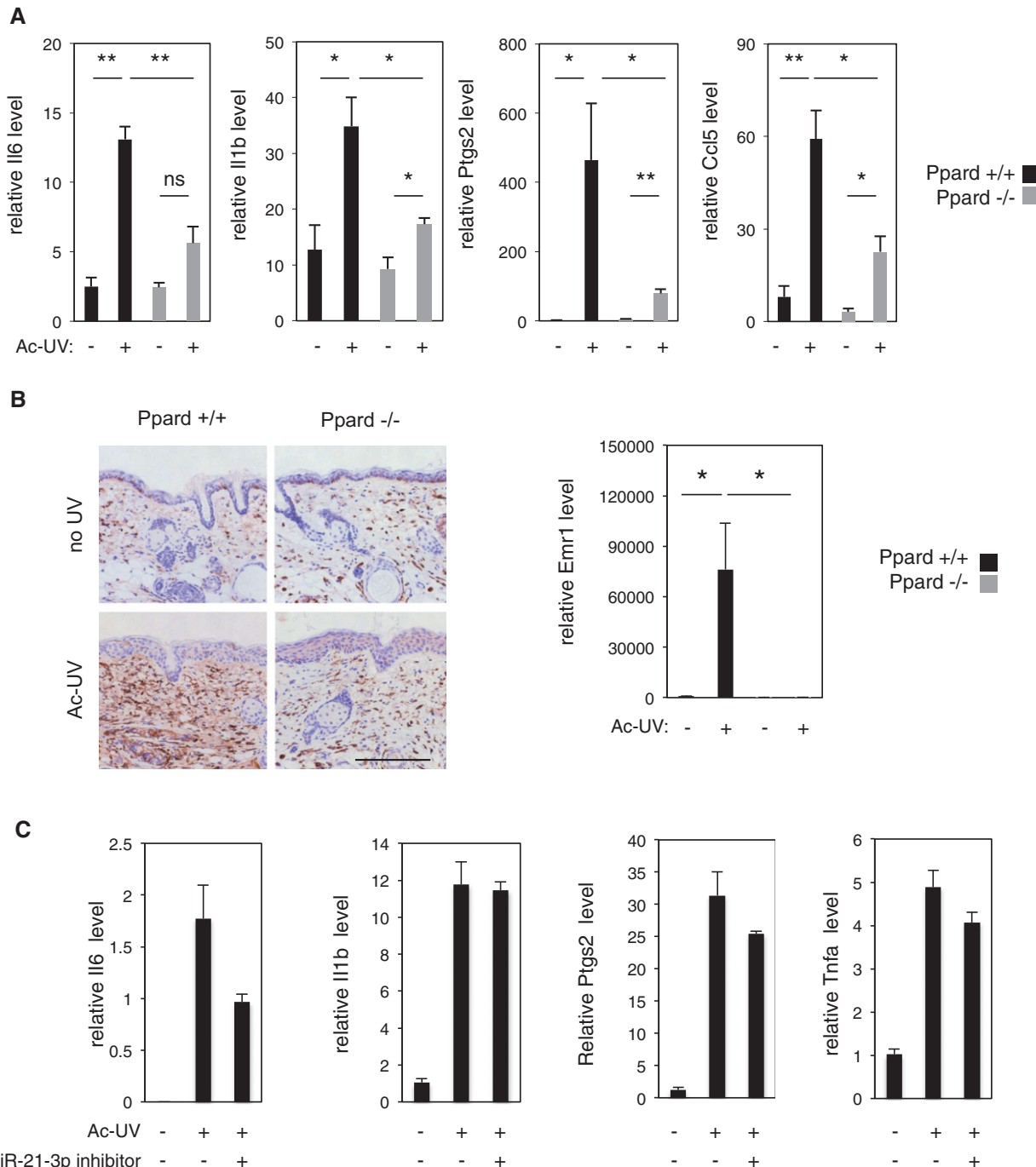

**Figure 5.  Elevated miR-21-3p level correlates with exacerbated inflammation in murine skin exposed to acute UV.**

A    RT–qPCR quantification of Il6, Il1b, Ptgs2 and Ccl5 levels in the epidermis of acutely irradiated (Ac-UV; +) and non-irradiated (−) Ppard[+/+] and Ppard[−/−] mice. Il6:
Ppard[+/+] no UV vs. Ac-UV $P = 7E-04$, Ppard[+/+] Ac-UV vs. Ppard[−/−] Ac-UV $P = 0.008$; Il1b: Ppard[+/+] no UV vs. Ac-UV $P = 0.010$, Ppard[+/+] Ac-UV vs. Ppard[−/−] Ac-UV
$P = 0.016$, Ppard[−/−] no UV vs. Ac-UV $P = 0.015$; Ptgs2: Ppard[+/+] no UV vs. Ac-UV $P = 0.028$, Ppard[+/+] Ac-UV vs. Ppard[−/−] Ac-UV $P = 0.028$, Ppard[−/−] no UV vs. Ac-UV
$P = 8E-04$; Ccl5: Ppard[+/+] no UV vs. Ac-UV $P = 0.002$, Ppard[+/+] Ac-UV vs. Ppard[−/−] Ac-UV $P = 0.014$, Ppard[−/−] no UV vs. Ac-UV $P = 0.010$; ns: non-significant. $N = 4$
animals per group, one representative experiment is shown out of three independent replicates.

B    Left panel: Immunostaining for infiltrating F4/80-positive macrophages in acutely irradiated (Ac-UV) and non-irradiated (no UV) Ppard[+/+] and Ppard[−/−] mice skin
sections. Scale bar: 200 μm. Right panel: RT–qPCR quantification of relative level of the macrophage marker Emr1 mRNA in total skin of acutely irradiated (Ac-UV; +)
and non-irradiated (−) Ppard[+/+] and Ppard[−/−] mice. Emr: Ppard[+/+] no UV vs. Ac-UV $P = 0.034$; Ppard[+/+] Ac-UV vs. Ppard[−/−] Ac-UV $P = 0.033$. $N = 4$ animals per group,
one representative experiment is shown out of three independent replicates.

C    RT–qPCR quantification of relative levels of Il6, Il1b, Ptgs2, and Tnfa in mouse epidermis acutely irradiated (Ac-UV; +) and non-irradiated (−), treated with miR-21-3p
inhibitor (+) or mismatched control (−) as indicated.

Data information: Results are presented as mean values ± SEM. The statistical comparison between groups was performed by using *t*-test. **P*-value < 0.05; ***P*-value < 0.01.

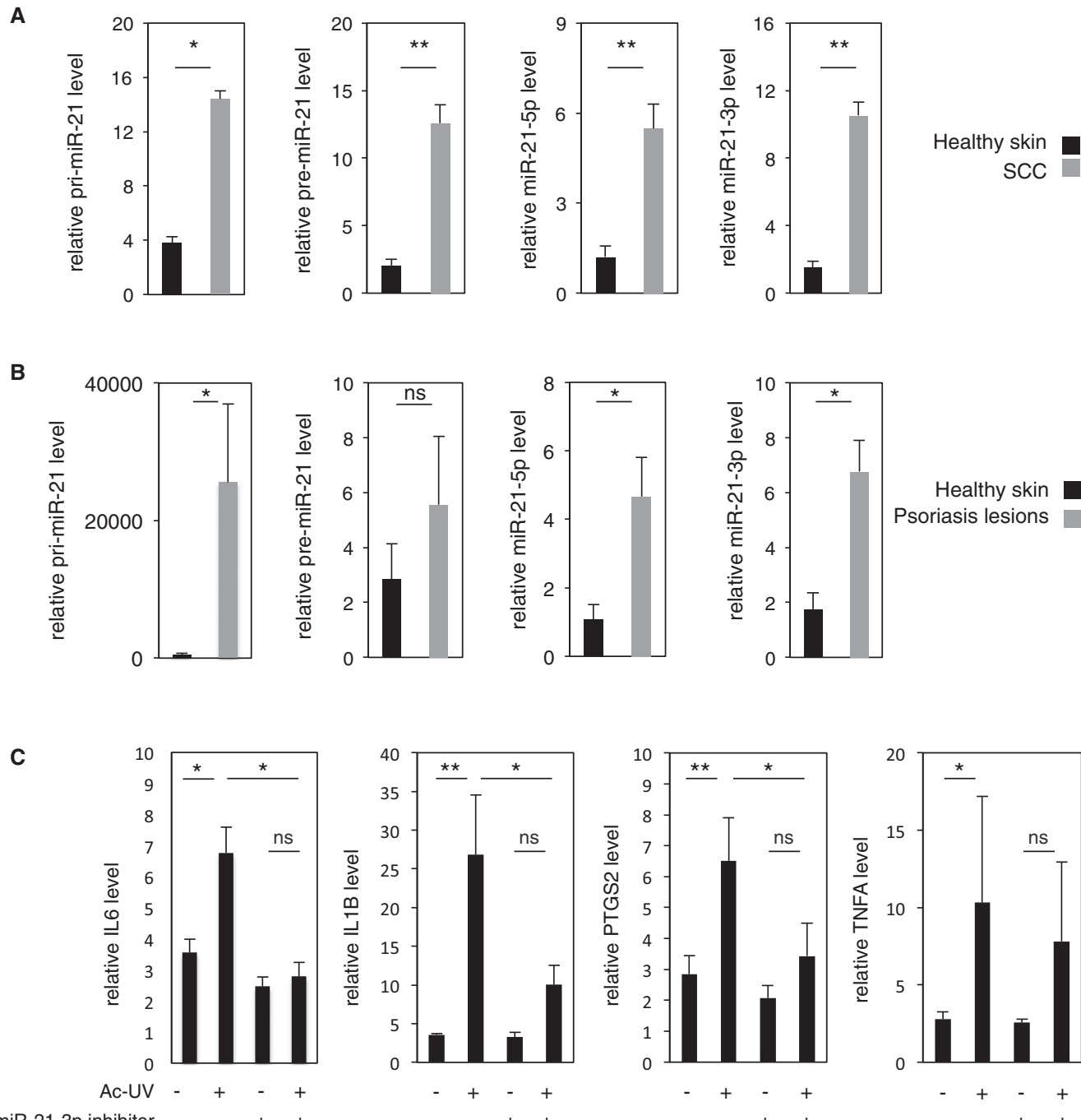

**Figure 6.  Elevated miR-21-3p and its inhibition are of clinical relevance to human skin inflammatory disorders.**

A   RT–qPCR quantification of relative levels of pri-miR-21, pre-miR-21, miR-21-5p, and miR-21-3p levels in healthy human skin and human squamous cell carcinomas (SCC). pri-miR-21 healthy skin vs. SCC $P = 0.043$; pre-miR-21 healthy skin vs. SCC $P = 3E-05$; miR-21-5p healthy skin vs. SCC $P = 7E-05$; miR-21-3p healthy skin vs. SCC $P = 1.4E-06$. $n \geq 5$ independent biopsies per condition.

B   RT–qPCR quantification of relative levels of pri-miR-21, pre-miR-21, miR-21-5p, and miR-21-3p levels in healthy human skin and human psoriasis lesions. pri-miR-21 healthy skin vs. psoriasis $P = 0.034$; miR-21-5p healthy skin vs. psoriasis $P = 0.034$; miR-21-3p healthy skin vs. psoriasis $P = 0.036$; ns: non-significant, $n \geq 4$ independent biopsies per condition.

C   RT–qPCR quantification of relative levels of IL6, IL1B, PTGS2, and TNFA levels in epidermis of abdominal healthy human skin biopsies exposed to acute UV *ex vivo*, with or without topical treatment with miR-21-3p inhibitor (+) or mismatched control (−) as indicated. ns: not significant. IL6: no UV vs. Ac-UV $P = 0.036$, Ac-UV vs. Ac-UV/miR-21-3p inhibitor $P = 0.019$; IL1B: no UV vs. Ac-UV $P = 0.002$, Ac-UV vs. Ac-UV/miR-21-3p inhibitor $P = 0.023$; PTGS2: no UV vs. Ac-UV $P = 0.008$, Ac-UV vs. Ac-UV/miR-21-3p inhibitor $P = 0.022$; TNFA: no UV vs. Ac-UV $P = 0.032$, ns = non-significant. $n \geq 7$ replicates, a pool of three independent experiments performed with the skin of three different donors is shown.

Data information: Results are presented as mean values ± SEM. The statistical comparison between groups was performed by using *t*-test. *$P$-value < 0.05; **$P$-value < 0.01.

compared to healthy tissue. Moreover, we show that topical administration of a miR-21-3p inhibitor on *ex vivo* human skin significantly reduces UV-induced inflammation. Thus, this study identifies miR-21-3p as a putative potent therapeutic target in the treatment of cutaneous inflammatory diseases.

Although miR-21-5p (commonly named miR-21) plays several recognized functions in the skin (Guo *et al*, 2009; Ma *et al*, 2011; Xu *et al*, 2012; Hou *et al*, 2013), its passenger miR-21-3p has attracted little attention in this organ to date, most likely due to its low basal expression. Contrary to the view that only one miRNA strand is functional, our study provides evidence that miR-21-3p expression is increased in keratinocytes upon various pathophysiological conditions, where it acts as a functional miRNA.

Using a combination of genetic and pharmacological approaches, we unravel that the remarkable increase of miR-21-3p in the epidermis exposed to UV involves the activation of PPARβ/δ and of TGFβ1 signaling. While the regulation of pri-miR21 level by PPARβ/δ is rather modest and likely involves other unidentified actors, the upregulation of mature miR-21-3p provoked by UV *in vivo* requires PPARβ/δ. These data lead us to suggest that PPARβ/δ-dependent regulation of miR-21-3p levels relies on combined activation of the transcription of pri-miR-21, reinforced downstream by post-transcriptional activation of its processing into mature miR-21-3p. The combination of PPARβ/δ and TGFβ1 gain- and loss-of-function indicates that the PPARβ/δ-dependent regulation of pri-miR-21 and miR-21-3p levels is indirect and requires TGFβ receptor activation. In line with our hypothesis that PPARβ/δ regulates miR-21-3p at the transcriptional and post-transcriptional levels, TGFβ was also shown to activate the transcription of pri-miR-21 as well as its processing to mature miR-21-5p (Davis *et al*, 2008; Godwin *et al*, 2010; Zhong *et al*, 2011). The identification of the PPARβ/δ/TGFβ1/miR-21-3p cascade provides a mechanism for miR-21-3p activation in response to UV, and it further suggests that miR-21-3p is a novel mediator of PPARβ/δ and TGFβ1 activities. PPARβ/δ and TGFβ1 are known to play critical roles in the maintenance of skin homeostasis (Di-Poi *et al*, 2004), in skin inflammation and skin carcinogenesis (Glick *et al*, 2008; Han *et al*, 2012; Montagner *et al*, 2013; Ravindran *et al*, 2014). Given that PPARβ/δ and TGFβ1 functions are, by far, not limited to the epidermis, a broader involvement of miR-21-3p in mediating their functions remains open and requires further investigation.

The canonical mode of miRNA action involves the downregulation of multiple targets, either by stimulating mRNA degradation or by inhibiting mRNA translation. We identify human SMAD7 as a direct target of miR-21-3p. However, following UV irradiation, the mouse Smad7 expression level does not anti-correlate with that of miR-21-3p in the epidermis, but correlates with that of TGFβ and IL-1β, in line with the fact that Smad7 is a direct TGFβ target gene (Nakao *et al*, 1997; Stopa *et al*, 2000), also known to be upregulated by IL-1β (Bitzer *et al*, 2000). Although miR-21-3p likely contributes to the regulation of its target at the post-transcriptional level, Smad7 is under unsurprising complex regulations *in vivo*. Besides SMAD7, *in silico* analyses predicted a whole repertoire of targets for miR-21-3p, in line with the hypothesis that miR-21-3p pro-inflammatory role most likely relies on the complex regulation of multiple mRNA targets, whose study goes beyond the scope of this manuscript.

Although frequently viewed as an anti-inflammatory transcription factor (Michalik & Wahli, 2007), PPARβ/δ regulation of

inflammation is complex and is context dependent. Previous work has shown that its activation with an agonist decreased inflammatory symptoms in mouse models of contact dermatitis (Plager *et al*, 2007; Kim *et al*, 2014), and cutaneous inflammatory response after 12-O-tetradecanoylphorbol-13-acetate (TPA) treatment was exacerbated in PPARβ/δ-deficient mice (Man *et al*, 2008). In contrast to these anti-inflammatory actions, PPARβ/δ increased expression in murine epidermis provoked a psoriasis-like phenotype, and its expression is elevated in human psoriatic lesions (Appendix Fig S2F), where it was proposed to contribute to the persistence of activated T cells (al Yacoub *et al*, 2008; Romanowska *et al*, 2010). Like PPARβ/δ, TGFβ1 exhibits a context-dependent regulation of skin inflammation (Li *et al*, 2004; Glick *et al*, 2008; Han *et al*, 2012). Of note, it was recently demonstrated to promote UV-induced cutaneous inflammation and tumor formation (Glick *et al*, 2008; Ravindran *et al*, 2014) and to activate the expression of miR-31, another keratinocyte pro-inflammatory miRNA of relevance in psoriasis (Xu *et al*, 2013). In line with a pro-inflammatory role of PPARβ/δ in response to UV, we show that the skin of Ppard$^{+/+}$ mice displays exacerbated inflammation in response to UV compared to those of Ppard$^{-/-}$ animals. Our finding that miR-21-3p is activated by the PPARβ/δ/TGFβ1 axis provides a mechanism for their pro-inflammatory actions in the skin. Moreover, our present report raises the hypothesis that the procarcinogenic role we described for PPARβ/δ in the murine skin exposed to UV (Montagner *et al*, 2013) is also mediated by higher chronic inflammation, to which contributes activation of miR-21-3p. In line with our study, and while the present manuscript was in revision, Ge and collaborators demonstrated that miR-21-3p was involved in skin squamous cell carcinoma progression (Ge *et al*, 2015). Whether the pro-inflammatory role of miR-21-3p extends to other tissues remains to be explored. In support of a broader pathophysiological role of miR-21-3p, it was recently described as a fibroblast-derived paracrine mediator that promotes cardiomyocyte hypertrophy (Bang *et al*, 2014) and its expression was reported in inflamed aortic endothelial cells (Bang *et al*, 2014), epithelial and hematolymphoid cancers, and sarcomas (Aure *et al*, 2013; Ge *et al*, 2015).

From a preclinical perspective, we demonstrate that topical inhibition of miR-21-3p in *ex vivo* human skin explants is sufficient to significantly reduce the epidermal response to acute UV exposure. Surprisingly, while few recent data suggested that miRNAs may be involved in the response to UV exposure in isolated cell cultures (Guo *et al*, 2009; Pothof *et al*, 2009; Dziunycz *et al*, 2010), their involvement in the global response of human skin to UV has not been addressed. Our data not only reveal miR-21-3p as an inflammatory mediator in human skin response to UV, but they also suggest that miR-21-3p inhibition is of therapeutic interest as an anti-inflammatory molecule in this context. Targeting miRNA for anti-cancer therapeutic purposes has been successful in various murine models (reviewed in Kasinski & Slack 2011). In these models, miRNA-based therapy demonstrated promising benefit with limited side effects. In the skin, delivery of miR-483-3p or miR-203 was shown to reduce the growth of subcutaneous squamous cell carcinoma (Bertero *et al*, 2013) or of basal cell carcinoma (Sonkoly *et al*, 2012), respectively, while silencing miR-21-5p provided therapeutic benefit in a preclinical model of patient-derived psoriatic lesions transplanted in immunocompromised mice (Guinea-Viniegra

*et al*, 2014). Here, we propose that silencing of miR-21-3p, as a single or combined therapy, may be of relevance to protect against inflammatory dermatoses caused by solar radiation, especially in the case of UV hypersensibility pathologies and UV-based phototherapy. Finally, as numerous miRNAs were either shown or predicted to regulate inflammation and cytokines, non-invasive topical targeting of miRNA is a valuable approach to explore to improve treatment of skin inflammatory disorders, which remain a significant therapeutic challenge. Topical delivery of miRNA-based therapy will likely decrease the risk for side effects, although further research is required to develop safe RNA chemistry and delivering agents able to cross the epidermal barrier.

# Materials and Methods

### Animal experimentation

All experiments involving animals were approved by the Veterinary Office of the Canton Vaud (Switzerland) in accordance with the Federal Swiss Veterinary Office Guidelines and conform to the Commission Directive 86/609/EEC. Mice were raised and housed in a standard colony (4–5 animals per cage), in a temperature- and light-controlled environment (12/12-h light/dark cycle) and maintained with water and food *ad libitum*. Animals were hairless albino SKH-1/Ppard$^{+/+}$ and SKH-1/Ppard$^{-/-}$ females, 8–12 week old, maintained on a mixed genetic background (SKH1-C57/Bl6-SV129). Ppard$^{+/+}$ and Ppard$^{-/-}$ were littermates. Each experiment involved four animals per group, randomly assigned to treatments, and was repeated independently. Skin samples were all collected from euthanized mice. *UV irradiation*: Briefly, mice from different litters were exposed to a single dose of 120 mJ/cm$^2$ of UVB for acute UV exposure, or to a dose of 70 mJ/cm$^2$ of UVB three times a week during a maximum period of 30 weeks as described for chronic exposure (Montagner *et al*, 2013). Non-irradiated (sham manipulated) aged-matched mice were used as controls. Total RNA containing the small RNA fraction was isolated from control skins, skins 24 h after acute UV exposure, lesion-free skins after 12 weeks of chronic exposure, papilloma, and SCCs biopsies. *PPARβ/δ inhibition*: Two hundred microliters of GSK0660 (Sigma; 625 μg/μl in 70% ethanol) was applied topically on the back of the animals, 1 h prior to UV exposure. *TGFβ1 inhibition*: Hundred microliters of the receptor kinase inhibitor SB431542 (Axon Medchem; 10 mM in DMSO) was topically applied on the back of the animals 15 min before and after acute exposure. Skin was then harvested and processed for dermis/epidermis separation 24 h after UV exposure. *In vivo miR-21-3p inhibition*: miR-21-3p inhibitor (fluorescently labeled *in vivo* miRCURY LNA™ Exiqon) or mismatch control in water was injected subcutaneously (10 mg/kg) 6 h before and 1 h following acute UV irradiation. Skin was harvested and processed for dermis/epidermis separation 24 h after UV exposure. *Tape stripping*: Back skin of hairless mice was stripped 10 times with ordinary adhesive tape. For each stripping, a fresh piece of tape was lightly pressed onto the skin and pulled off. Twenty-four hours after tape stripping, the skin was harvested and processed for dermis/epidermis separation. *DMBA/TPA carcinogenesis*: Six-week-old C57BL/6J female mice (*N* = 45) were subjected to DMBA two-hit multistage skin carcinogenesis protocol as previously described (Owens *et al*,

1999). Mice were shaved 2 days before initiation and regularly during the protocol. Mice were topically treated with 200 nmol of DMBA in 0.2 ml acetone. One week later, mice were treated twice a week for 6 weeks with 5 nmol PMA (phorbol 12-myristate 13-acetate) in 0.2 ml acetone. A second hit of DMBA was performed on the eighth week followed by the resumption of PMA treatment for 14 more weeks as described above. Control mice were topically treated with 0.2 ml acetone vehicle. Mice were monitored weekly for papilloma and tumors counting. Control skin (acetone treated), hyperplastic skin, papilloma, and tumors were harvested throughout the protocol, and biopsies were frozen for RNA extractions.

### Epidermis and dermis separation

Epidermis and dermis were prepared from whole dorsal skin of mice according to the protocol developed by Clemmensen *et al* (2009). After dorsal skin harvest, adipose tissue was removed with a scalpel on ice; the skin was cut into thin strips (1–2 mm in large) and immediately incubated for 15 min at RT in ammonium thiocyanate (3.8% in PBS 1×). Epidermis and dermis were then separated mechanically with forceps and scalpel. Reverse transcription followed by qPCR was performed for specific epidermal (keratins 10 and 14) and dermal (collagen 14α1) markers to check for proper separation of the two compartments.

### MiRNA *in situ* hybridization with Locked nucleic acid (LNA) probes

Skin samples were fixed and cryopreserved at 4°C in 4% in paraformaldehyde and 30% sucrose overnight. Samples were then frozen in optimum cutting temperature matrix (OCT) at −80°C, further cut into 10-μm sections (Cryostat, Leica), and mounted on slides. During the acetylation step, sections were washed in 0.1 M of triethanolamine/10 min and then in triethanolamine/0.25% acetic anhydride/10 min. Hundred microliters of digoxigenin-labeled LNA probes (Exiqon, 25 nM in hybridization buffer) was added on each section, and hybridization was performed O/N (16 h) at 54°C (20°C below probes melting temperature) in a humid chamber. Following hybridization, sections were washed three times with 0.2× SCC/20 min/60°C and equilibrated in TN buffer (100 mM Tris–HCl pH 7.5, 150 mM NaCl) for 5 min at room temperature. Sections were then blocked in TNB buffer (0.5% blocking reagent, PerkinElmer in TN buffer) for 30 min at room temperature. To quench peroxidase activity, the slides were incubated in 3% H$_2$O$_2$ for 1 h and washed with TNT (0.05% Tween-20 in TN buffer) 3 × 5 min. Diluted anti-DIG-POD 1:500 in TNB was applied and incubated for 30 min at room temperature. After three washes with TNT 3 × 5 min, slides were incubated 10 min at room temperature with diluted Cy3-tyramide 1:50 in amplification reagent (TSATM Plus Cy5 Fluorescence System, PerkinElmer). Finally, slides were washed with TNT 3 × 5 min and mounted in Mowiol.
LNA probes were designed by Exiqon:
mmu-miR-21-3p:    5DigN/GACAGCCCATCGACTGCACTGCTGTTG/3Dig_N/

has-miR-21-3p: 5DigN/CAACACCAGUCGAUGGGCUGU/3Dig_N/
Scramble-miR:/5DigN/GTGTAACACGTCTATACGCCA/3Dig_N/

## Western blot for SMAD7 and Smad7

Anti-human and mouse SMAD7/Smad7 antibodies were purchased from Abcam (catalogue number ab124890). Anti-human and anti-mouse GAPDH antibody was from Cell Signaling Technologies (catalogue number 2118). Ten micrograms (HaCat cells) or 20 µg (mouse epidermis) of proteins was separated on SDS–polyacrylamide gels. Proteins were then transferred onto a polyvinylidene filter in semidry conditions (PerfectBlue peqlab). Membranes were incubated overnight with anti-SMAD7/Smad7 primary antibody (1:1,000) in 5% BSA and then 1 h with anti-rabbit IgG-HRP antibody (1:30,000; Promega) in 1% BSA. GAPDH was used as a loading control (1:10,000; 1% BSA). Signals were detected using Western Bright Quantum (Advansta) and Fusion FX (Vilber Lourmat) and quantified with the Bio1D software (Vilber Lourmat).

## F4/80 macrophages detection

Immunohistochemical detection of the macrophage marker F4/80 was performed on hairless mice skin paraffin sections using rat anti-F4/80 primary antibody (1:800) (Abcam, catalogue number ab6640) incubated overnight at 4°C in 2.5% NGS buffer and a goat anti-rat HRP as a secondary antibody (Biosource, catalogue number AR 13404) incubated during 40 min in 2.5% NGS buffer at room temperature.

## Patient SCC and psoriasis biopsies

Cutaneous SCC samples were obtained anonymously from the Department of Dermatology, University Hospital of Lausanne, Switzerland. Normal skin was from healthy adult volunteers or from the edges of skin tumors. Experienced pathologists diagnosed SCC biopsies.

Cutaneous psoriasis samples were obtained anonymously from the Department of Dermatology, Universitäts Spital Zürich, Switzerland. Psoriasis samples were obtained from adult volunteers and diagnosed by experienced pathologists. Informed consent for research was obtained prior to routine diagnostic services.

## Human skin *ex vivo* explant culture and miR-21-3p inhibitor topical application

Abdominal skin biopsies were obtained anonymously from the Department of Internal Medicine, University Hospital of Geneva, Switzerland, and from the Department of Musculoskeletal Medicine Biobank, University Hospital of Lausanne, Switzerland. Informed consent for research was obtained prior to surgery. For those samples obtained from Department of Musculoskeletal Medicine Biobank, informed consent was regulated through the Department of Musculoskeletal Medicine Biobank (University Hospital of Lausanne, Switzerland) with Profs. Raffoul and Applegate. Immediately after surgery, human abdominal skin was cleaned from its adipose tissue and 0.5 mm$^2$ biopsies were deposited on adjusted agar-based bed-filled dishes and incubated in humidified incubator at 37°C with 5% $CO_2$ for 24 h. Topical application of the miR-21-3p-based lotion was performed 24 and 2 h before UV irradiation (120 mJ/cm$^2$), immediately and 2 h after UV irradiation. Skin was processed for dermis–epidermis separation prior to RNA extraction 18–24 h after UV irradiation. *Agar-based bed-filled dishes*: 0.25 g of agar powder was

diluted in 10 ml of sterilized water and heated for 1 min (microwave). One milliliter of agar mix was added for each 10 ml of DMEM GlutaMAX-I media (Gibco), 20% of FBS, and 1% of antibiotic (penicillin/streptomycin). Six milliliters of this solution was quickly added to 6-cm petri dishes. When solidified, a "bed" was dug in the agar in which skin biopsy was deposed, epidermis remaining at surface. *MiR-21-3p-based lotion*: A 9:1:1:8 mix of glycerol:DMSO: transfection reagent (RNAimax, Invitrogen):50 µM of miR-21-3p LNA inhibitor (Exiqon, I-has-miR-21-3p) or its control (Exiqon, MM-has-miR-21-3p) diluted in PBS was prepared. Sixteen microliters of this miR-21-3p antagonist-based lotion was applied to each 0.5 mm$^2$ biopsies, 24 and 2 h before UV irradiation, immediately, 2 and 24 h after UV irradiation. Skin was processed for dermis–epidermis separation prior to RNA extraction 24 h after UV irradiation.

## Cell culture, transfection, and treatments

Human immortalized keratinocytes HaCaT (Cell Line Service; Germany) cells were maintained in DMEM growth medium (Gibco) supplemented with 4,500 mg/l glucose, 10% fetal bovine serum, 100 units/ml of penicillin G, and 100 µg/ml of streptomycin. Cells were grown in a 5% $CO_2$ atmosphere at 37°C. Assignment to various treatments was randomized. *PPARβ/δ activation*: PPARβ/δ agonists GW0742 or GW501516 in DMSO were added at a final concentration of 0.5 µM, and cells were harvested after 24 h of treatment. *miR-21-3p gain of function*: miR-21-3p mimic (miRIDIAN, Thermo Scientific Dharmacon) or cel-miR-67-3p used as a control (miRIDIAN, Thermo Scientific Dharmacon) was transfected to the cells at a concentration of 50 nM using the Lipofectamine RNAimax (Invitrogen). Cells were harvested 24 h after transfection for gene expression analysis. *TGFβ1 treatment*: Cells were grown for 24 h in the presence of 2 ng/ml or 5 ng/ml of human TGFβ1 (reconstituted in 5 mM HCl, Prospec) or vehicle (PBS containing 2 mg/ml of albumin). *Combined TGFβ1 inhibition and PPARβ/δ activation*: TGFβ type I receptor kinase (ALK5) inhibitor SB431542 (Axon Medchem) 2 µM in DMSO was added to the cells for 24 h. The PPARβ/δ activator GW0742 was then added to cells at a final concentration of 0.5 µM in DMSO. Cells were harvested 24 h later. *Cycloheximide treatment*: HaCaT cells were treated for 6 h with vehicle (DMSO) or PPARβ/δ agonist GW0742 (5 µM) in the presence or in the absence of cycloheximide (1 µg/ml).

## Luciferase assays

Wild-type SMAD7 3′UTR was purchased from OriGene (Clone NM_005904). Mutant SMAD7 3′UTR was obtained by PCR mutagenesis of the wild-type SMAD7 3′UTR, followed by sequencing for quality control (Fig 3B). Mycoplasm-free (MycoAlert, Lonza) human embryonic kidney (HEK) 293 cells (gift from Prof. W. Herr, University of Lausanne) were maintained in DMEM + 10% FBS. HEK 293 cells were transfected (Lipofectamine RNAimax) with pRL for standardization (Renilla luciferase reporter under CMV promoter, 0.625 ng/ml), miR-21-3p mimic or scrambled sequence (miRIDIAN, Thermo Scientific Dharmacon, 50 nM), and Firefly luciferase reporter—wild-type or mutant Smad7 3′UTR reporter (40 ng/ml). Cells were harvested 24 h after transfection for luciferase assay analysis (Dual-Glo® Reporter Assay System, Promega) using the Microplate luminometer GloMax®-9 (Promega).

## Extraction of total RNA, reverse transcription, and real-time PCR

Total RNA was isolated using TRIzol reagent (Invitrogen). Quality of the RNA was assessed using a BioAnalyzer (Agilent). One microgram of total RNA was reverse-transcribed with random hexamer primers (Promega) using the SuperScriptII Reverse Transcriptase (Invitrogen). Real-time PCR was performed with SYBR Green PCR Master Mix (Applied Biosystems) using Stratagene Mx3000P thermo cycler. mRNA expression was related to the expression of the housekeeping genes GAPDH and Ef1a1 for murine samples and to RPL27 and HPRT1 or ACTB for human samples and cells (GeNorm, M value < 0.5). For miRNA, reverse transcription was performed using the Exiqon miRCURY LNA™ Universal RT microRNA kit. Quantitative PCR was performed according to manufacturer protocol with microRNA LNA™ PCR primer sets and SYBR Green PCR Master Mix (Applied Biosystems) on a Stratagene Mx3000P thermo cycler. MiR-103 and sno234 were used as control for normalization.

Expression of mature mmu-miR-21-3p and mmu-miR-21-5p in control skin (acetone treated), DMBA/TPA-treated hyperplastic skin, papilloma, and tumors was evaluated using TaqMan MicroRNA Assays (Applied Biosystems) and the Lightcycler 480 detection system (Roche Applied Science, Indianapolis, IN, USA). Expression levels were normalized to mmu-SnoR55.

Quantification of relative expression was based on the determination of the threshold cycle (Ct).

### mRNA primers

| Name | Forward | Reverse |
|---|---|---|
| mmu-GAPDH | GTATGACTCCACTACGGCAAA | TTCCCATTCTCGGCTTG |
| mmu-Eef1a1 | CCTGGCAAGCCCATGTGT | TCATGTCACGAACAGCAAAGC |
| mmu-Ppard | CGGCAGCCTCAACATGG | AGATCCGATCGCACTTCTCATAC |
| hsa-PPARD | GCATGAAGCTGGAGTACGAGAAG | GCATCCGACCAAAACGGATA |
| hsa-SERPINE1 | GGCTGACTTCACGAGTCTTTCA | ATGCGGGCTGAGACTATGACA |
| hsa-p21 (CDKN1A) | CTCAGAGGAGGCGCCATGTCAGAAC | GCAGCCCGCCATTAGCGCAT |
| hsa-RUNX | TCGCTTTCAAGGTGGTGGCCC | GCGGTAGCATTTCTCAGCTCAGCC |
| hsa-SNAI2 | AGCGAACTGGACACACATACAGTGAT | GCGTGGAATGGAGCAGCGGTA |

ANGPTL/Angptl4, TGFB1/Tgfb1, RPL27/Rpl27, HPRT1/Hprt, SMAD7/Smad7, Emr1, CASP14/Casp14, CCL5/Ccl5, CXCL10/Cxcl10, IL1B/Il1b, IL1RAP/Il1rap, PTGS2/Ptgs2, IL6/Il6, TNFA/Tnfa, COL4A1/Col4a1, KRT5/Krt5, and IVL/Ivl have been purchased from Qiagen (QuantiTect primers).

### miRNA primers

mmu/hsa-miR-21-3p, mmu/hsa-miR-21-5p, mmu/hsa-miR-103, and a customized hsa- and mmu-pre-miR-21 primers have been purchased from Exiqon (microRNA LNA PCR primer sets).

hsa-pri-miR-21: forward-TTTTGTTTTGCTTGGGAGGA; reverse AGCAGACAGTCAGGCAGGAT.

mmu-pri-miR-21: forward-CCAGAGATGTTTGCTTTGCTT; reverse-TGCCATGAGATTCAACAG.

### mRNA microarray data analysis/pathway analysis

All RNA quantities were assessed by NanoDrop®ND-1000 spectrophotometer and the RNA quality was assessed using RNA 6000 NanoChips with the Agilent 2100 Bioanalyzer (Agilent, Palo Alto, USA). For each sample, 100 ng of total RNA was amplified using the Ambion® WT Expression Kit (4411973, Life Technologies) kit. 5.5 μg of the cDNA was fragmented and labeled with GeneChip® WT Terminal Labeling kit (901525, Affymetrix). Affymetrix mouse gene 1.0ST arrays (Affymetrix, Santa Clara, CA, USA) were hybridized with 2.3 μg of fragmented target, at 45°C for 16 h washed and stained according to the protocol described in Affymetrix GeneChip® Expression Analysis Manual (Fluidics protocol FS450_0007). Background subtraction, quantile normalization, and probeset summarization were performed with the *Affymetrix Power Tools* command line tool, using the RMA method. Normalized chip data were stored as ExpressionSet objects in R using the Bioconductor *affy* and *affyPLM* packages. Differentially expressed genes between conditions were detected by applying the empirical Bayes method (*limma* package in R; Gentleman *et al*, 2004; Smyth, 2004) and ranked according to fold change. Four biological replicates for each group (non-irradiated mice, 12 weeks irradiated mice or miR-21-3p mimic and control-treated HaCaT cells) were used for the differential expression analysis. *P*-values were adjusted for multiple comparisons using the Benjamini–Hochberg procedure (Benjamini & Hochberg, 1995), and genes with an adjusted *P*-value of < 0.01 were considered as differentially expressed. Gene enrichment analysis has been realized with DAVID version 6.7 (the Database for Annotation, Visualization and Integrated Discovery) identifying enriched biological GO terms (da Huang *et al*, 2009).

### miRNA microarray data analysis

Each RNA skin sample was prepared according to the Agilent's miRNA Microarray System protocol and loaded on the mouse microarray (Mouse miRNA Microarray Release 16.0). Normalization procedures were based on the invariant procedure and the quantile normalization as described in Pradervand *et al* (2009). After normalization, the limma package in R was used to define a linear regression for inter-conditions fold change calculation as for the microarray analysis. *P*-values were adjusted for multiple comparisons using the Benjamini–Hochberg procedure (Benjamini & Hochberg, 1995), and miRNAs with a *P*-value of < 0.001 and absolute fold change of at least 1.5 were selected as differentially regulated.

### RNA-Seq (miRNA) in mouse skin

High-quality RNA were extracted from the 16 dermis and epidermis samples of hairless Ppard$^{+/+}$ and Ppard$^{-/-}$ mice acutely irradiated or not irradiated (control) using TRIzol standard extractions. Small RNA libraries were prepared using 1 μg of total RNA according to the TruSeq Small RNA Sample Preparation Guide (Illumina, San Diego, CA). Libraries were sequenced either on the Illumina HiSeq 2000 (26 samples) or on the HiSeq 2500 (44 samples) using v3 chemistry.

Formatting and processing of sequence reads: Sequence reads were processed to remove the adaptor sequences and reformatted to FASTA files using the FASTX-Toolkit (http://hannonlab.cshl.edu/fastx_toolkit/index.html). To generate count data, the raw sequences were compared to mouse mature microRNA sequences (from miRBase version 17) and non-coding RNA sequences (Rfam version 10) using MEGABLAST with a word size of 8 nucleotides. The criteria for counting a sequence match were if the % query was $\geq 90\%$ of the target sequence and if there were $\leq 2$ mismatches over the alignment. The % query was calculated as $(a/q) \times p$ where $a$ = alignment length, $q$ = query length, and $p$ = percent identity over aligned region. The matches against miRBase were parsed and the top matches (based on % query) were selected. If a sequence had more than one top match against different database sequences, it was excluded from the subsequent analysis. Matches to Rfam were only taken into account for sequences not matching miRBase. Statistical analysis of microRNA counts: Counts for mouse mature microRNAs were used to test for differential expression. Count data from Rfam microRNAs were excluded since these sequences may represent non-mature microRNAs. To perform the analysis, count data were fitted to a statistical model based on the negative binomial distribution using the R package DESeq, which is useful for detecting significant RNA-Seq variation with a low number of biological replicates (Anders & Huber, 2010). To perform the normalization and differential expression analysis, raw read counts per gene were normalized to the relative size of each library. Empirical dispersion (the squared coefficient of variance for each gene) was estimated using the *pooled* method. Here, samples from all conditions with replicates are used to estimate a single pooled dispersion value, which is applied to all samples. The dispersion–mean relationship was then fitted using the *local* method and the *maximum* of the fitted and empirical values was used in subsequent calculations. The difference between the means of KO vs. WT samples was then calculated using a negative binomial test and *P*-values were adjusted for multiple comparisons using the Benjamini–Hochberg method (Benjamini & Hochberg, 1995). miRNA with an adjusted *P*-value of $\leq 0.05$ was considered to be differentially expressed.

### RNA-Seq reads of miR-21-3p in mouse brain, heart, kidney, and testis tissues

RNA-Seq reads of miR-21-3p were obtained from high-quality RNA of mouse brain, heart, kidney, and testis samples. Small RNA-seq libraries and sequencing protocols are described in (Meunier *et al*, 2013).

### Statistical analyses

Results are presented as mean values ± standard error of mean (SEM). Unless mentioned otherwise, the statistical comparison between groups was performed by using *t*-test, a maximum of three comparisons were performed per panel, and robustness of statistical significance was verified after correction for multiple testing. Probability was considered to be significant at $P < 0.05$.

### The paper explained

#### Problem

The skin offers effective permeability barrier and provides protection against harmful conditions. Excessive UV exposure disrupts this barrier, provoking inflammation, premature aging, and ultimately, UV-induced carcinogenesis. Beyond UV-induced inflammation, skin disorders accompanied by acute or chronic inflammation are the most common dermatological pathologies. They remain a significant therapeutic challenge and represent a heavy personal burden as well as enormous costs for society. Despite the prevalence of these diseases, the complexity of the underlying molecular mechanisms remains incompletely characterized. Further studies of these mechanisms will provide novel and alternative therapeutic targets to explore in order to improve treatment of skin inflammatory disorders.

The nuclear hormone receptor peroxisome proliferator-activated receptor β/δ (PPARβ/δ) regulates inflammation and promotes skin healing following an injury, through activation of keratinocyte proliferation, survival, and migration. In contrast to these beneficial functions, we recently showed that activation of PPARβ/δ also favored the progression of UV-induced skin squamous cell carcinoma in mouse. In order to get further insights into the mechanisms underlying these functions of PPARβ/δ, we explored miRNA that would orchestrate the PPARβ/δ-dependent skin response to UV exposure. While miRNA are widely recognized as important players in skin homeostasis and functions, their involvement in skin response to UV has not been addressed.

#### Results

We used SKH-1 hairless albino mice lacking PPARβ/δ, human keratinocytes, and human skin explants in culture, which are all relevant model for studying skin inflammatory responses. We unveil that upon UV exposure, PPARβ/δ activates the expression of the passenger miRNA miR-21-3p. We identify miR-21-3p as a novel UV-induced miRNA in keratinocytes. We establish that miR-21-3p upregulation by PPARβ/δ in response to UV is indirect and requires activation of the TGFβ1 receptor. We further show that miR-21-3p plays a pro-inflammatory role in keratinocytes and that its increase in human skin is of pathophysiological importance, as we found elevated levels of miR-21-3p in human skin exposed to UV, in human squamous cell carcinoma and human psoriatic lesions. Importantly, we provide evidence that inhibition of miR-21-3p reduces cutaneous inflammation in *ex vivo* human skin biopsies exposed to UV.

#### Impact

We describe a novel molecular cascade that participates in the inflammatory skin response to excessive UV exposure and that is likely involved in the procarcinogenic role played by PPARβ/δ in the murine skin exposed to UV. This cascade is of broader pathophysiological importance, since PPARβ/δ and miR-21-3p levels are not only increased in murine and human skin exposed to UV, but also in human psoriatic skin and in human squamous cell carcinomas compared to healthy tissue. From a preclinical perspective, we identify miR-21-3p as an important target to explore for cutaneous anti-inflammatory treatment. Our data motivate the evaluation of topical delivery of miRNA as a valuable approach to explore to improve treatment of skin inflammatory disorders.

### Data availability

Primary data: Data access: NCBI Gene Expression Omnibus (GEO) (http://www.ncbi.nlm.nih.gov/geo/) accession number GSE80431.
Referenced data:
Meunier *et al* (2013)

NCBI Gene Expression Omnibus (GEO) (http://www.ncbi.nlm.nih.gov/geo/) accession number GSE40499.

Expanded View for this article is available online.

## Acknowledgements

The authors wish to acknowledge the Genomic Technologies Facility of the University of Lausanne for microarray analyses and RNA sequencing; Dr. S. Pradervand and Dr. J. Weber for guidance in microarrays analysis; F. Lammers for help with the construction of the mutant SMAD7 3′UTR; and M. Baruchet, N. Fares, C. Tallichet-Blanc, and C. Moret for excellent technical support. We wish to thank Dr. G. Ponzio for DMBA/TPA-treated murine skin samples, Prof. W. Herr for helpful discussion and support and Dr. GP. Rando and Dr. S. Vollers for critical reading of the manuscript. We thank Prof. W. Raffoul and Prof. L. Applegate, who are responsible for the Department of Musculoskeletal Medicine Biobank (University Hospital Lausanne), and kindly provided us with patient tissue samples. We are grateful to Prof. W. Wahli for a long-lasting fruitful cooperation and his continued support. This study was supported by FNRS Sinergia grant (26073695 CRSI33-130576 grant to Dr. L. Michalik and Prof. W. Wahli), University of Lausanne Herbette and UNIL Foundations (grants 26078145; 26075125; 26078149 to Dr. L. Michalik), and by the Etat de Vaud, InnoPACTT (grant to Dr. G. Degueurce).

## Author contributions

GD, IDE, CP, AM, JM, RR, and GP designed and performed experiments and analyzed the data. FS analyzed the data and performed statistical analyses. NCB, DH, AK, PJ, and GH collected, classified, and provided patient biopsies of healthy skin, SCC, and psoriasis lesions. MI analyzed RNA sequencing data. IX designed microarray experiments. AB, LMu, and MS performed experiments. LMi supervised the project, designed experiments, and analyzed the data. GD and LMi wrote this manuscript. All authors reviewed the manuscript.

## Conflict of interest

The authors declare that they have no conflict of interest.

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
