## [Review Process File · EMBO Molecular Medicine]

Identification of a novel PPAR β/δ / miR-21-3p axis in UV-induced skin inflammation

Gwendoline Degueurce, Ilenia D'Errico, Christine Pich, Mark Ibberson, Frédéric Schütz, Alexandra Montagner, Marie Sgandurra, Lionel Mury, Paris Jafari, Akash Boda, Julieneunier, Roger Rezzonico, Nicolò Costantino Brembilla, Daniel Hohl, Antonios Kolios Günther Hofbauer, Ioannis Xenarios, Liliane Michalik

Corresponding author: Liliane Michalik, University of Lausanne

Review timeline:

Submission date:	27 April 2015
Editorial Decision:	10 June 2015
Revision received:	08 March 2016
Editorial Decision:	07 April 2016
Revision received:	21 April 2016
Accepted:	27 April 2016

Transaction Report:

Editor: Céline Carret

1st Editorial Decision

10 June 2015

Thank you for the submission of your manuscript to EMBO Molecular Medicine. I apologise for the unusual delay in getting back to you, for some unclear reasons, we experienced difficulties in securing three expert and willing reviewers. We have now heard back from the three referees whom we asked to evaluate your manuscript. Although the referees find the study to be of potential interest, they also raise a number of concerns that need addressing in the next final version of your article.

You will see from the comments below that the three referees find the study timely, relevant and convincing. Nevertheless, they also suggest a few additional experiments to strengthen the findings and improve conclusiveness. Of importance for our scope, the additional *in vivo* data suggested by both referees 1 and 2 should be addressed, along with missing controls, explanations and better statistical analyses to be provided.

Given these evaluations, I would like to give you the opportunity to revise your manuscript, with the understanding that the referees' concerns must be fully addressed and that acceptance of the manuscript would entail a second round of review. Please note that it is EMBO Molecular Medicine policy to allow only a single round of revision and that, as acceptance or rejection of the manuscript will depend on another round of review, your responses should be as complete as possible.

Please read below for important editorial formatting.

I look forward to receiving your revised manuscript.

***** Reviewer's comments *****

Referee #1 (Comments on Novelty/Model System):

Important controls are missing, see my comments to the authors. Without those the claims are not supported by the provided evidence.

Referee #1 (Remarks):

Degueurce and colleagues described the UV-dependent induction of mir-21-3p in vivo in mice and ex vivo in human keratinocytes. They found that this induction was PPAR / and TGF -dependent. They identified SMAD7 as a direct target gene of mir-21a-3p as well as a pro-inflammatory activity of mir-21a-3p in mouse and human skin. The topic is relevant and interesting, but critical control experiments are missing to allow the authors to draw conclusions.

Major points of criticism:

1. The authors showed that mir-21-3p was regulated at transcription level in human keratinocytes by PPAR / and TGF . It is important to show that pri-mir-21 level is also regulated PPAR -dependent manner in vivo in mouse epidermis.
2. The authors suggested that SMAD7 is a direct target of mir-21-3p, but they did not test it using luciferase reporter system following the cloning of wild-type and mutant mir-21-3p binding-sites. It is important for the demonstration of direct miRNA-mRNA interaction.
3. The authors showed that SMAD7 was repressed in keratinocytes at protein level by overexpression of mir-21-3p mimic. Is important to show that SMAD7 expression at protein level is correlated with mir-21-3p level in vivo in UV-treated wild-type and PPAR / -deficient mice.

4. The authors suggested that mir-21-3p could enhance TGF signalling activity. However, they did not show whether TGF -target genes can be expressed at higher level in mir-21-3p mimic-overexpressing cells following TGF treatment.

5. The authors showed that mir-21-3p has a pro-inflammatory activity in keratinocytes. What is the potential mechanism of this action? Is it depend on mir-21-3p-repressed SMAD7 expression?

Referee #2 (Remarks):

This manuscript addresses the role of miRNA-21-3p in UV induced skin inflammation. The authors identify miRNA-21-3p as a target of PPARb/d during UV stimulation of keratinocytes and skin. They show that UV treated epidermis requires the activation of TGFb-1 by PPARb/d. The data are convincing that this molecular pathway is important for inflammation induced by keratinocytes. While other groups have implicated other mir-21 isoforms in psoriasis and skin inflammation, this manuscript nicely ties this family to PPAR and TGFb signaling. By addressing a few comments below, the manuscript should be appropriate for EMBO Mol. Med.

1. The title seems inaccurate. The authors do not show that targeting the mir-21-3p results in decreased inflammation in the skin. They do show that inflammatory genes are decreased (Figure 6C). A more accurate title would be replace "Targeting the" with "Identifying a"

2. The statistics used are sometimes not appropriate. In general, the authors use T-tests. However, when more than 2 groups are analyzed, one-way ANOVA is more appropriate. Also, in Figure 2A, the significance between control and GW0742 is hard to believe given the variability in the GW0742 treated sample.

3. The authors utilize a miR-21-3p inhibitor and show that UV induction of inflammatory cytokines is reduced. The authors should analyze TGF-b1 since their model suggests that it is downstream of miR-21-3p. It would be nice to see how the inhibitor alters UV induced inflammation in vivo.

Minor comments:

1. There is a typo in Figure 2C y axis.
2. In Figure 2, labeling B with TGFb1 would be helpful for readers.

Referee #3 (Comments on Novelty/Model System):

The authors employ strong models for testing their hypothesis. They utilised knockout mouse models of PPARd to examine the role of this nuclear receptor in mir-21-3p inflammatory signalling. They extend their findings beyond murine models and examine the effectors of mir21-3p in human keratinocytes and human skin explants to make translational observations.

Referee #3 (Remarks):

The study entitled, "Targeting the novel PPARb/d/mir21-3p axis to reduce UV-induced skin inflammation" by Degueurce et al. examines the regulatory role of PPAR b/d in the downstream signalling of mir21-3p within the context of inflammatory response during UV-induced skin damage

and additional skin maladies including squamous cell carcinomas. The study is timely and of great interest. It would benefit from addressing the following points:

- 1) Figure 1a only displays the ratio of mir-21-5p to mir-21-3p. It would be useful to see the total expression of mir21-3p in the skin compared to the other tissues to help ascertain its relative importance. Also please provide the mir-21-5p levels to confirm the change in ratio is not driven exclusively by decreases in mir-21-5p.
- 2) Page 6, the authors claim, "in vivo topical inhibition of PPAR δ with an antagonist reduced the magnitude of mir-21-3p UV dependent increase in the skin of Ppard $+/+$ mice." However the levels of mir 21-3p +/- the ppar δ antagonist do not appear to be significantly different (Fig 1e). Can the authors address this?
- 3) The authors describe SMAD7 as a putative direct target of mir-21-3p (page7). What additional targets were identified by their in silico analysis?
- 4) Fig 5b represents just PGE2 and PGE3 pro-inflammatory eicosanoids. However the LC-MS method described (Le Faouder et al 2013) measures an extensive range of both pro-inflammatory and inflammatory-resolving lipid species. Given PPAR δ 's known role in anti-inflammatory processes, it is important to describe any additional species of eicosanoid changing to gain a more in-depth understanding of the inflammatory profile.
- 5) Fig 6c page 10. It is clear that the increase in the pro-inflammatory factors as a result of UV exposure is dampened by mir-21-3p inhibition. However, it is not clear whether the increase in IL6, IL1B, PTGS2 and TNFA are significant in the skin explants treated with both UV and the mir-21-3p inhibitor compared to those just treated with the inhibitor.
- 6) Did mir-21-3p inhibition (Fig 6c) affect the downstream inflammatory factors described in figure 5b (PGE2 + PGE3).

Minor:

- 1) reference required page 4, first sentence. "Although PPARs were recently reported to regulate expression of some miRNAs..."
- 2) Please provide p-value cut off for supplementary table 1.
- 3) replace "twice more" with "double the" page 9.
- 4)page 12 "Pparb" is used whereas "Ppard" is used elsewhere. This must be consistent.
- 5) page 13 remove the superfluous "side" from the final sentence of discussion.

Point-by-point reply to the reviewers' remarks

Referee #1 (Comments on Novelty/Model System):

Important controls are missing, see my comments to the authors. Without those the claims are not supported by the provided evidence.

Referee #1 (Remarks):

Degueurce and colleagues described the UV-dependent induction of mir-21-3p in vivo in mice and ex vivo in human keratinocytes. They found that this induction was PPAR β and TGF β -dependent. They identified SMAD7 as a direct target gene of mir-21a-3p as well as a pro-inflammatory activity of mir-21a-3p in mouse and human skin. The topic is relevant and interesting, but critical control experiments are missing to allow the authors to draw conclusions.

Major points of criticism:

1. The authors showed that mir-21-3p was regulated at transcription level in human keratinocytes by PPAR β and TGF β . It is important to show that pri-mir-21 level is also regulated PPAR β -dependent manner in vivo in mouse epidermis.

Answer:

Quantification of pri-miR-21 expression levels was performed in the epidermis of Ppard $+/+$ and $-/-$ mice, as suggested by the reviewer. The data shown in revised Figure 1c (left panel) indicate that UV-induced up-regulation of pri-miR-21 is significantly reduced in Ppard $-/-$ compared to Ppard $+/+$ epidermis (p value 0.022), confirming transcriptional PPAR β/δ -dependent regulation of pri-miR-21.

This novel data is described on Page 6.

2. The authors suggested that SMAD7 is a direct target of mir-21-3p, but they did not test it using luciferase reporter system following the cloning of wild-type and mutant mir-21-3p binding-sites. It is important for the demonstration of direct miRNA-mRNA interaction.

Answer:

Luciferase reporter assays were performed to compare the activity of a wild-type and a mutated SMAD7 3'UTR reporter, co-transfected with a miR-21-3p mimic or a mismatched control sequence. The mutated SMAD7 3'UTR was obtained by introducing three mutations in each of the two miR-21-3p predicted binding site, as indicated in revised Figure 3b (left panel).

Revised Figure 3b (right panel) shows that the luciferase activity of the wild-type SMAD7 3'UTR reporter was reduced by 44% by the delivery of the miR-21-3p mimic, while the luciferase activity of the mutant SMAD7 3'UTR reporter was unaffected. These novel data are described and discussed on Page 8 and show that SMAD7 is a direct target of miR-21-3p.

3. The authors showed that SMAD7 was repressed in keratinocytes at protein level by overexpression of mir-21-3p mimic. Is important to show that SMAD7 expression at protein level is correlated with mir-21-3p level in vivo in UV-treated wild-type and PPARb-deficient mice.

Answer:

Western blots were performed on proteins extracted from Ppard +/+ and -/- epidermis, non-irradiated or following acute exposure to UV, to quantify the level of endogenous Smad7. Smad7 was below detection level in the epidermis of non-irradiated animals (data not shown), as also described by He and colleagues (*He et al, Oncogene (2001) 20, 471-483*).

Following UV irradiation, Smad7 expression level increased to a greater extent in Ppard+/+ compared to Ppard-/- epidermis, as shown in revised Figure 3e. Thus, although Smad7 is a bona fide target of miR-21-3p (Figure 3a-c), its expression level does not anti-correlate with that of miR-21-3p in UV-irradiated mouse epidermis *in vivo* (Figure 1e). This indicates that the expression of Smad7 in the epidermis *in vivo* is the result of unsurprising complex regulations. Notably, *in vivo* expression of Smad7 in the epidermis (Figure 3e) correlates with that of TGF β (Figure 2d) and IL1b (Figure 5a), consistent with the fact that Smad7 is a direct TGF β target gene (*Nakao et al., Nature (1997) 389, 631-635* ; *Stopa et al, J. Biol. Chem (2000) 275(38), 29308-17*), also known to be up regulated by IL1b (*Bitzer et al, Genes and Dev, (2000) 14(2), 187-97*).

These novel data, described and discussed on Page 8 (revised Figure 3e) do not rule out that miR-21-3p contributes to the regulation of Smad7 expression *in vivo*, but suggest that other regulators, such as TGF β and IL1b, have a greater impact than miR-21-3p on Smad7 levels in the epidermis exposed to UV.

4. The authors suggested that mir-21-3p could enhance TGFb signaling activity. However, they did not show whether TGFb target genes can be expressed at higher level in mir-21-3p mimic-overexpressing cells following TGFb treatment.

Answer:

The expression levels of several TGF β -1 target genes were quantified by RT-qPCR in miR-21-3p mimic-overexpressing HaCat cells following TGF β -1 treatment.

Consistent with an activation of the TGF β pathway by miR-21-3p, the TGF β -1 target genes SERPINE, p21 and RUNX were expressed at a significantly higher level in miR-21-3p mimic-overexpressing cells following TGF β -1 treatment. mRNA levels of TGFB1 itself or its target SNAI2 were not significantly affected by TGF β -1 or miR-21-3p treatments. These novel data, shown in revised Figure 3d and described on Page 8 indicate that miR-21-3p significantly enhances TGF β -1 signaling.

5. The authors showed that mir-21-3p has a pro-inflammatory activity in keratinocytes. What is the potential mechanism of this action? Is it dependent on mir-21-3p-repressed SMAD7 expression?

Answer:

We fully agree with the reviewer that studying the mechanism of miR-21-3p pro-inflammatory action is of interest. miR-21-3p pro-inflammatory action may indeed be partially mediated by its repression of SMAD7. However, as miRNA activity is known to be the result of multiple mRNA and/or protein regulations, it is unlikely that SMAD7 is the only mediator of miR-21-3p pro-inflammatory action. This is now addressed in the Discussion. Due to the complexity of miRNA mode of action, we think that the investigation of this mechanism goes beyond the scope of the present manuscript and we prefer not to address this complex issue here experimentally.

Referee #2 (Remarks):

This manuscript addresses the role of miRNA-21-3p in UV induced skin inflammation. The authors identify miRNA-21-3p as a target of PPAR β /d during UV stimulation of keratinocytes and skin. They show that UV treated epidermis requires the activation of TGF β -1 by PPAR β /d. The data are convincing that this molecular pathway is important for inflammation induced by keratinocytes. While other groups have implicated other mir-21 isoforms in psoriasis and skin inflammation, this manuscript nicely ties this family to PPAR and TGF β signaling. By addressing a few comments below, the manuscript should be appropriate for EMBO Mol. Med.

1. The title seems inaccurate. The authors do not show that targeting the mir-21-3p results in decreased inflammation in the skin. They do show that inflammatory genes are decreased (Figure 6C). A more accurate title would be replace "Targeting the" with "Identifying a"

Answer:

The title has been modified according to the reviewer's suggestion. Our manuscript is now entitled "Identification of a novel PPAR β / δ / miR-21-3p axis in UV-induced skin inflammation".

2. The statistics used are sometimes not appropriate. In general, the authors use T-tests. However, when more than 2 groups are analyzed, one-way ANOVA is more appropriate. Also, in Figure 2A, the significance between control and GW0742 is hard to believe given the variability in the GW0742 treated sample.

Answer:

To properly answer the reviewer's concern, the statistics were reviewed in collaboration with a professional statistician. On each panel, a maximum of three comparisons were performed. The data are highly significant, and remained significant after correction for multiple testing. These precisions have been added in the Materials and Methods section.

Figure 2a: the p value obtained for the comparison between control and GW0742 treatments is 0.0094, confirming that GW0742 treatment had a significant impact on pri-miR-21 expression level in HaCat cells.

3. The authors utilize a miR-21-3p inhibitor and show that UV induction of inflammatory cytokines is reduced. The authors should analyze TGF- β 1 since their model suggests that it is downstream of miR-21-3p. It would be nice to see how the inhibitor alters UV induced inflammation in vivo.

Answer:

We have analyzed TGFB1 expression level in human skin biopsies treated with miR-21-3p inhibitor as suggested by the reviewer. TGFB1 mRNA level was not affected by the miR-21-3p inhibitor, as shown in revised Appendix Figure 2h, indicating no impact of miR-21-3p on TGFB1 expression itself. These data, combined with our novel data showing that mimic delivery in HaCat cells did not affect TGFB1 either (revised Figure 3d), suggest that miR-21-3p does not regulate the expression of TGFB1 itself, but TGF β -1 signaling. These data are described on Page 11 respectively

Following the reviewer suggestion, we have performed *in vivo* delivery of miR-21-3p mimic, with the limitations imposed by the veterinary regulation. These novel data are shown in revised Figure 5c and described on Page 10. Data indicate a clear anti-inflammatory trend of miR-21-3p delivery, which notably reduced Il6, Ptgs2 and TNF α UV-induced expression, although the calculated p value comparing UV-induced Il6, Ptgs2 and TNF α levels with or without miR-21-3p treatment did not reach statistical significance.

Minor comments:

1. There is a typo in Figure 2C y axis.

Answer:

Figures were all scrutinized for typos

2. In Figure 2, labeling B with TGFb1 would be helpful for readers.

Answer:

Figure 2b has been labeled with TGF β -1 according to the reviewer's suggestion.

Referee #3 (Comments on Novelty/Model System):

The authors employ strong models for testing their hypothesis. They utilised knockout mouse models of PPAR δ to examine the role of this nuclear receptor in mir-21-3p inflammatory signalling. They extend their findings beyond murine models and examine the effectors of mir21-3-p in human keratinocytes and human skin explants to make translational observations.

Referee #3 (Remarks):

The study entitled, "Targeting the novel PPAR δ /mir21-3p axis to reduce UV-induced skin inflammation" by Degueurce et al. examines the regulatory role of PPAR δ in the downstream signalling of mir21-3p within the context of inflammatory response during UV-induced skin damage and additional skin maladies including squamous cell carcinomas. The study is timely and of great interest. It would benefit from addressing the following points:

1) Figure 1a only displays the ratio of mir-21-5p to mir-21-3p. It would be useful to see the total expression of mir21-3p in the skin compared to the other tissues to help ascertain its relative importance. Also please provide the mir-21-5p levels to confirm the change in ratio is not driven exclusively by decreases in mir-21-5p.

Answer:

Relative expression levels of miR-21-3p and miR-21-5p were quantified in mouse brain, heart, kidney and epidermis. Data are presented in revised Figure 1a, in replacement of the miR-21-3p/miR-21-5p ratio (now shown in revised Appendix Figure 1a). These novel data indicate enrichment of miR-21-3p in the epidermis compared to other tissues, which is not driven by a lower expression of miR-21-5p. These data are described on Page 5.

2) Page 6, the authors claim, "in vivo topical inhibition of PPAR δ with an antagonist reduced the magnitude of mir-21-3p UV dependent increase in the skin of Ppard $+/+$ mice." However the levels of mir 21-3p \pm the ppard antagonist do not appear to be significantly different (Fig 1e). Can the authors address this?

Answer :

Statistical analyses were reviewed by a professional statistician (see also answer to the second reviewer, comment n°2). The data presented in Revised Figure 1e

show a significant increase of miR-21-3p expression level in the epidermis of Ppard +/+ mice (p value = 0.0019), which is significantly reduced by topical treatment of mouse dorsal skin with the PPAR β antagonist GSK0660 (p value 0.0392)

3) The authors describe SMAD7 as a putative direct target of mir-21-3p (page7). What additional targets were identified by their in silico analysis?

Answer:

An Appendix table is now provided (Appendix Table 2), which shows the complete list of putative miR-21-3p targets identified in our *in silico* analysis of miRNA Recognition Elements using the miRmap interface (Vejnar et al, 2013). We refer to Appendix table 2 on Page 8.

4) Fig 5b represents just PGE2 and PGE3 pro-inflammatory eicosanoids. However the LC-MS method described (Le Faouder et al 2013) measures an extensive range of both pro-inflammatory and inflammatory-resolving lipid species. Given PPARd's known role in anti-inflammatory processes, it is important to describe any additional species of eicosanoid changing to gain a more in-depth understanding of the inflammatory profile.

Answer:

Upon careful re-examination of the *in vivo* eicosanoid analysis, we think that these data are not reliable. As they represent a minor part of our study and are not central to the description of the role of miR-21-3p and its activation by Ppard and UV, we have decided to remove these data from the manuscript.

5) Fig 6c page 10. It is clear that the increase in the pro-inflammatory factors as a result of UV exposure is dampened by mir-21-3p inhibition. However, it is not clear whether the increase in IL6, IL1B, PTGS2 and TNFA are significant in the skin explants treated with both UV and the mir-21-3p inhibitor compared to those just treated with the inhibitor.

Answer:

While the human skin explants responded slightly to UV in the presence of mir-21-3p inhibitor, the increase in IL6, IL1B, PTGS2 and TNFA was not significant compared to the levels quantified in skin explants treated with the inhibitor only. The calculated p values (“UV + miR-21-3p inhibitor” compared to “no UV + miR-21-3p inhibitor”) are as follows:

- IL6: p=0.69
- IL1B: p=0.057
- PTGS2: p= 0.29
- TNFA: p=0.69.

Figure 6c has been modified to clarify this information: on revised Figure 6c, "ns" clearly indicates non-significant statistical analyses.

6) Did mir-21-3p inhibition (Fig 6c) affect the downstream inflammatory factors described in figure 5b (PGE2 + PGE3).

Answer:

As mentioned in our answer to the reviewer comment number 4, we believe that our eicosanoid analyses *in vivo* is not fully reliable. As these data represent a minor part of our study and are not central to the description of the role of miR-21-3p and its activation by Ppard and UV, and we have decided to remove these data from the manuscript, and thus eicosanoid analyses in human skin explants are no longer relevant.

Minor:

1) reference required page 4, first sentence. "Although PPARs were recently reported to regulate expression of some miRNAs..."

Answer:

References were added according to the reviewer suggestion.

2) Please provide p-value cut off for Appendix table 1.

Answer:

Statistical significance was set to p-value < 0.001 (t-test analysis adjusted for multiple comparison). The information is now available in Appendix table 1 and the corresponding legend.

3) replace "twice more" with "double the" page 9.

Answer:

We have corrected the sentence according to the reviewer suggestion.

4)page 12 "Pparb" is used whereas "Ppard" is used elsewhere. This must be consistent.

Answer:

We have corrected this mistake according to the reviewer suggestion.

5) page 13 remove the superfluous "side" from the final sentence of discussion.

Answer:

We have corrected this mistake according to the reviewer suggestion.

Thank you for the submission of your revised manuscript to EMBO Molecular Medicine. We have now received the enclosed reports from the referees that were asked to re-assess it.

As you will see while one referee is now fully supportive, referee 1 remains concerned about the in vivo data. Following further editorial advice, we decided to move forward with acceptance, providing that you can thoroughly discuss your findings in the lines highlighted by this referee. Please make sure to provide a point-by-point response and make the changes in the manuscript visible (ideally, providing 2 versions: 1 without track changes, and 1 with).

Please submit your revised manuscript within two weeks. I look forward to seeing a revised form of your manuscript as soon as possible.

***** Reviewer's comments *****

Referee #1 (Comments on Novelty/Model System):

The mechanisms uncovered are only partially support the claims and there are many other factor at play unaccounted for. See detailed comments to authors.

Referee #1 (Remarks):

The authors made an effort to address the issues raised and provide new data. However not all the concerns are addressed and specially the in vivo data they provide or not provide (i.e. miRNA 21-3p SMAD anti-correlation) point into a much more complex situation than described and claimed. They now show that the identified miRNA miR21-3p is able to inhibit SMAD7 and they also provide in vivo evidence that pro-miRNA 31-p is regulated in vivo although to a much less impressive way as the mature miRNA. In fact their data suggest that there are other powerful (yet unidentified) regulators acting in vivo. Therefore the in vivo PPARb/d dependence is partial at best. In addition it remains doubtful how relevant their findings are in vivo and especially in a disease state. In other words the identified components of a regulatory circuit are novel, but the in vivo situation is clearly more complex than the title is suggesting and the mechanisms claimed are partial at best. The analyses provided are not sufficiently comprehensive to account for the many variables contributing to the effects. These facts significantly reduce the enthusiasm of the reviewer.

Referee #3 (Comments on Novelty/Model System):

The authors have addressed my previous concerns and the manuscript is now, in my opinion, suitable for publication.

Referee #3 (Remarks):

The authors have addressed my previous concerns and the amendments made have improved the quality of the paper.

Point-by-point reply to the reviewers' remarks

Referee #1 (Remarks):

The authors made an effort to address the issues raised and provide new data. However not all the concerns are addressed and specially the *in vivo* data they provide or not provide (i.e. miRNA 21-3p SMAD anti-correlation) point into a much more complex situation than described and claimed. They now show that the identified miRNA miR21-3p is able to inhibit SMAD7 and they also provide *in vivo* evidence that pro-miRNA 31-p is regulated *in vivo* although to a much less impressive way as the mature miRNA. In fact their data suggest that there are other powerful (yet unidentified) regulators acting *in vivo*. Therefore the *in vivo* PPAR β/δ dependence is partial at best. In addition it remains doubtful how relevant their findings are *in vivo* and especially in a disease state. In other words the identified components of a regulatory circuit are novel, but the *in vivo* situation is clearly more complex than the title is suggesting and the mechanisms claimed are partial at best. The analyses provided are not sufficiently comprehensive to account for the many variables contributing to the effects. These facts significantly reduce the enthusiasm of the reviewer.

Answer:

The skin response to UV, as well as skin diseases like psoriasis or Squamous Cell Carcinoma (SCC) are extremely complex pathophysiological situations. We thus agree with the reviewer that the PPAR β/δ -TGF β -miR-21-3p cascade is one cascade among the many regulatory events at play in these situations. In characterizing this PPAR β/δ -dependent molecular cascade, we do not exclude that other regulatory mechanisms are also involved *in vivo*. Along this line, we discussed Smad7 regulation in our first answer to the reviewer comments.

Getting a comprehensive understanding of this complexity *in vivo* and in human diseases is hardly possible, in spite of decades of intensive research in this field. Nevertheless, we believe that our study contributes to a better understanding of this complexity. Moreover, we provide evidence that despite the complexity of the skin response to UV, inhibiting miR-21-3p is sufficient to reduce UV-induced inflammation *in vivo* in mouse and in human skin explants *ex vivo*.

Unsurprisingly, the transcription of the gene encoding pri-miR-21 and miR-21-3p is under complex transcriptional regulation. For example, it is also regulated via AP1 (Fujita, J et al, J Mol Biol. 2008), which is a major family of TXF activated by UV. Our novel data show that the UV-induced expression of pri-miR-21 *in vivo* is partially but significantly reduced in the absence of PPAR β/δ (Figure 1C). This shows that, although other factors are involved, PPAR β/δ do contributes to the UV-induced upregulation of pri-miR-21 transcription. Besides the regulation of pri-miR-21, our data convincingly show that the UV-induced increase of the mature miR-21-3p *in vivo* is fully PPAR β/δ -dependent (Figure 1C; 1D; 1E), indicating that the PPAR β/δ -dependent regulation is reinforced at the level of the mature miR-21-3p. Collectively, these data lead us to suggest that PPAR β/δ -dependent regulation of miR-21-3p levels relies on two combined mechanisms: (i) transcriptional regulation of pri-miR-21 by PPAR β/δ (and other unidentified factors), (ii) further reinforced downstream by post-transcriptional regulation along the processing of pri-miR-21 to mature miR-21-3p.

Our experiments combining PPAR β/δ and TGF β activation or inhibition furthermore indicate that the PPAR β/δ -dependent regulation of pri-miR-21 and miR-21-3p requires TGF β

activation. In line with our hypothesis that PPAR β/δ regulates miR-21-3p levels at the transcriptional and post-transcriptional levels, TGF β was also shown to activate the transcription of pri-miR-21 as well as its processing to mature miR-21-5p (Davis et al, Nature, 2008 ; Godwin et al, PNAS 2010; Zhong et al, J Am. Soc. Neph.).

Our laboratory focuses on the functions of PPARs and is not equipped to study the complex maturation of mature miRNA. Although we agree that it is of high interest and relevance, studying the promoter of the miR-21-3p encoding gene and its global transcriptional regulation, or how PPAR β/δ regulates the various maturation steps of miR-21-3p, goes beyond the scope of the present manuscript.

As experiments cannot be performed to formally prove that the PPAR β/δ -TGF β -miR-21-3p cascade is involved in psoriasis or SCC in patients, our assumption that PPAR β/δ activation and miR-21-3p activation sustain disease states in patients relies on the combination of the following evidence:

- We showed that PPAR β/δ promotes SCC progression in mouse (Montagner et al, EMBO Molecular Medicine 2013), and Romanowska and colleagues showed that it promotes a psoriasis-like disease in mouse (Romanowska et al, PLoS One 2010),
- We provide evidence that miR-21-3p is proinflammatory in human keratinocytes (Figure 4), and Ge and colleagues described that it promotes SCC progression in mouse (Ge et al, Nature Cell Biology 2015; published while this manuscript was in revision),
- We show that high levels of PPAR β/δ and miR-21-3p correlate with SCC and psoriasis in patients (Figure 6 and Appendix Figure 2)
- We show that miR-21-3p inhibition in normal human skin *ex vivo* is sufficient to reduce UV-induced inflammation (Figure 6C). Moreover, we showed that inhibition of PPAR β/δ is also sufficient to reduce mouse skin response to UV (Montagner et al, EMBO Molecular Medicine 2013).

We believe that these are convergent evidence of the relevance of our data in disease states, although we agree that the formal proof that PPAR β/δ or miR-21-3p activation promotes diseases in patients is lacking.

We have completed the Discussion section of our manuscript to take the reviewer concern into consideration.

Referee #3 (Remarks):

The authors have addressed my previous concerns and the amendments made have improved the quality of the paper.

Answer :

We wish to thank the Reviewer for her/his positive comments. We thank the Reviewer for her/his consideration and we are glad that the she/he finds our work acceptable for publication

Point-by-point reply to the Editor's requests

1) indicate in legends exact p= values, not a range. Some people found that to keep the figures clear, providing a supplemental table with all exact p-values was preferable. You are welcome to do this if you want to.

Answer:

The exact P values are now included in the Figure legends.

2) Data of gene expression experiments described in submitted manuscripts should be deposited in a MIAME-compliant format with one of the public databases. We would therefore ask you to submit your microarray data to the ArrayExpress database maintained by the European Bioinformatics Institute for example. ArrayExpress allows authors to submit their data to a confidential section of the database, where they can be put on hold until the time of publication of the corresponding manuscript. Please see <http://www.ebi.ac.uk/arrayexpress/Submissions/> or contact the support team at arrayexpress@ebi.ac.uk for further information. RNAseq data should also be deposited in appropriate database and accession number(s) provided.

Please adjust the Author Checklist accordingly.

Our data are published in NCBI Gene Expression Omnibus (GEO)
(<http://www.ncbi.nlm.nih.gov/geo/>).

As recommended by NCBI (see below), a single accession number is provided in the Materials and Methods section, Data Availability, to access all our data.

* GSE80431 is the reference Series for your publication:

<http://www.ncbi.nlm.nih.gov/geo/query/acc.cgi?acc=GSE80431>

* This SuperSeries record provides access to all of your data and is the best accession to be quoted in any manuscript discussing the data.

The data are not publicly available yet, but will be released provided that our manuscript is accepted for publication in EMBO Molecular Medicine.

In line with the submission of these data, we now provide a more detailed description of the methods used to analyze the RNA-Seq (miRNA) in mouse skin.

3) Appendix file: please adjust labelling such as it reads everywhere: Appendix Table S1, Appendix Figure S1 and so on.

Answer:

Appendix files and legends labelling has been adjusted

4) Every published paper now includes a 'Synopsis' to further enhance discoverability. Synopses are displayed on the journal webpage and are freely accessible to all readers. They include a short stand first (maximum of 300 characters, including space) as well as 2-5 one sentence bullet points that summarise the paper. Please write the bullet points to summarise the key NEW findings. They should be designed to be complementary to the abstract - i.e. not repeat the same text. We encourage inclusion of key acronyms and quantitative information (maximum of 30 words / bullet point). Please use the passive voice. Please attach these in a separate file or send them by email, we will incorporate them accordingly.

Answer:

A synopsis has been included to our manuscript

5) As part of the EMBO Publications transparent editorial process initiative (see our Editorial at <http://embomolmed.embopress.org/content/2/9/329>), EMBO Molecular Medicine will publish online a Review Process File (RPF) to accompany accepted manuscripts.

In the event of acceptance, this file will be published in conjunction with your paper and will include the anonymous referee reports, your point-by-point response and all pertinent correspondence relating to the manuscript. If you do NOT want this file to be published or if you want to remove any figures from it prior to publication, please inform the editorial office at contact@embomolmed.org.

Answer:

As also mentioned in our cover letter to the editors, we wish to delay our decision until formal acceptance of our manuscript.

Corresponding Author Name: Liliane Michalik

Manuscript Number: EMM-2015-05384